# Meta-analysis of epigenome-wide association studies in neonates reveals widespread differential DNA methylation associated with birthweight

Leanne K. Küpers et al.[#]

Birthweight is associated with health outcomes across the life course, DNA methylation may be an underlying mechanism. In this meta-analysis of epigenome-wide association studies of 8,825 neonates from 24 birth cohorts in the Pregnancy And Childhood Epigenetics Consortium, we find that DNA methylation in neonatal blood is associated with birthweight at 914 sites, with a difference in birthweight ranging from −183 to 178 grams per 10% increase in methylation ($P_{Bonferroni} < 1.06 \times 10^{-7}$). In additional analyses in 7,278 participants, <1.3% of birthweight-associated differential methylation is also observed in childhood and adolescence, but not adulthood. Birthweight-related CpGs overlap with some Bonferroni-significant CpGs that were previously reported to be related to maternal smoking (55/914, $p = 6.12 \times 10^{-74}$) and BMI in pregnancy (3/914, $p = 1.13 \times 10^{-3}$), but not with those related to folate levels in pregnancy. Whether the associations that we observe are causal or explained by confounding or fetal growth influencing DNA methylation (i.e. reverse causality) requires further research.

Correspondence and requests for materials should be addressed to D.A.L. (email: d.a.Lawlor@bristol.ac.uk) or to C.L.R. (email: caroline.relton@bristol.ac.uk) or to H.S. (email: h.snieder@umcg.nl) or to J.F.F. (email: j.felix@erasmusmc.nl). [#]A full list of authors and their affiliations appears at the end of the paper.

Intrauterine exposures, such as maternal smoking, pre-pregnancy body mass index (BMI), hyperglycaemia, hypertension, folate and famine are associated with fetal growth and hence birthweight[1–6]. Observational studies show that birthweight is also associated with later-life health outcomes, including cardio-metabolic and mental health, some cancers and mortality[7–11]. In these long-term associations, birthweight may act as a proxy for potential effects of intrauterine exposures[12,13]. Several mechanisms may explain the associations of intrauterine exposures with birthweight and later-life health as we illustrate in Fig. 1. Our overall conceptual framework in this study was that the intrauterine environment induces epigenetic alterations, which influence fetal growth and hence correlate with birthweight. This is partly supported by previous large-scale epigenome-wide association studies (EWAS) that have reported associations of relevant maternal pregnancy exposures, including smoking, air pollution and BMI, with DNA methylation in offspring neonatal blood[14–16]. However, whilst four previous EWAS have observed associations of DNA methylation with birthweight[17–20], the evidence to date has been limited in scale and power with sample sizes ranging from approximately 200 to 1000.

In this study, we hypothesised that there are associations between DNA methylation and birthweight. We further aimed to explore if these epigenetic alterations are associated with later disease outcomes (Fig. 1). If birthweight is a proxy for a range of adverse prenatal exposures, we might expect neonatal blood DNA methylation to be associated with birthweight. However, we acknowledge that any associations of DNA methylation with birthweight may be explained by confounding[21] or reflect fetal growth influencing DNA methylation.

Here we present a large meta-analysis of multiple EWAS to explore associations between neonatal blood DNA methylation and birthweight. In further analyses, we explore whether any birthweight-associated differential methylation persists at older ages. To aid functional interpretation, we (i) explore the overlap of identified cytosine-phosphate-guanine sites (CpGs) that are differentially methylated in relation to birthweight with those known to be associated with intrauterine exposure to smoking, famine and different levels of BMI and folate; (ii) associate DNA methylation at identified CpGs with gene expression and (iii) explore potential causal links with birthweight and later-life health using Mendelian randomization (MR)[22]. We show that DNA methylation in neonatal blood is associated with birthweight and some of the differential methylation is also observed in childhood and adolescence, but not in adulthood.

Also, we show overlap between birthweight-related CpGs and CpGs related to intrauterine exposures. Potential causality of the associations needs to be studied further.

## Results

**Participants.** We used data from 8825 neonates from 24 studies in the Pregnancy And Childhood Epigenetics (PACE) Consortium, representing mainly European, but also African and Hispanic ethnicities with similar proportions of males and females. Details of participants used in all analyses are presented in Table 1, Supplementary Data 1 and study-specific Supplementary Methods.

**Meta-analysis.** Primary, secondary and follow-up analyses are outlined in the study design in Fig. 2. Methylation at 8170 CpGs, measured in neonatal blood using the Illumina Infinium® Human-Methylation450 BeadChip assay and adjusted for cell-type heterogeneity[23–25], was associated with birthweight (false discovery rate (FDR) <0.05), of which 1029 located in or near 807 genes survived the more stringent Bonferroni correction ($p < 1.06 \times 10^{-7}$, Supplementary Data 2). We observed both positive (45%) and negative (55%) directions of associations between methylation levels of these 1029 CpGs and birthweight (Fig. 3) and these CpGs were spread throughout the genome (orange track (1) in Fig. 4 and Supplementary Fig. 1). We found evidence of between-study heterogeneity ($I^2 > 50\%$) for 115 of the 1029 sites (Supplementary Data 2), thus we prioritised 914 CpGs, located in or near 729 genes, based on $p < 1.06 \times 10^{-7}$ and $I^2 \leq 50\%$ for further analyses (Fig. 3 and orange track (1) in Fig. 4). The CpG with the largest positive association was cg06378491 (in the gene body of *MAP4K2*). For each 10% increase in methylation at this site, birthweight was 178 g higher (95% confidence interval (CI): 138, 218 g). The CpG with the largest negative association was cg10073091 (in the gene body of *DHCR24*), which showed a 183 g decrease in birthweight per 10% increase in methylation (95% CI: −225, −142 g). The CpG with the smallest *P*-value and $I^2 \leq 50\%$ was cg17714703 (in the gene body of *UHRF1*), which showed a 130 g increase in birthweight for 10% increase in methylation (95% CI: 109, 151 g).

Findings were consistent with results from our main analyses when restricted to participants of European ethnicity, with a Pearson correlation coefficient for effect estimates of 0.99 for the 914 birthweight-associated CpGs (Supplementary Fig. 2, blue track (2) in Fig. 4 and Supplementary Data 3) and 0.90 for all 450k CpGs. Comparing the main meta-analysis to the four Hispanic cohorts and

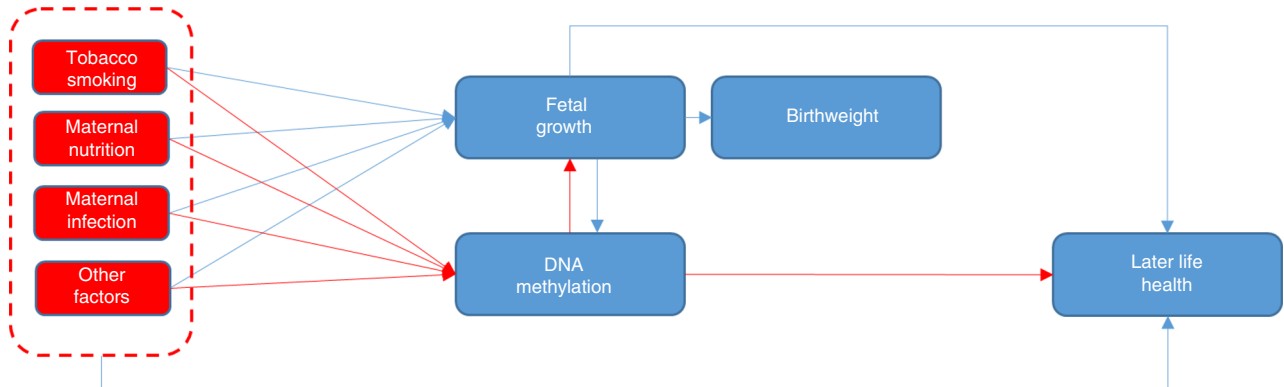

**Fig. 1** Hypothetical paths that might link intrauterine exposures to DNA methylation, birthweight and later-life health outcomes. Red arrows summarise the paths that have motivated the analyses undertaken in this study (i.e. that maternal environmental exposures influence DNA methylation that in turn influences fetal growth and hence birthweight). The EWAS meta-analysis undertaken sought to identify methylation associated with birthweight. Blue arrows summarise other plausible paths, including that maternal exposures influence fetal growth first and it then influences DNA methylation or that maternal exposures may influence fetal growth/birthweight and later-life health outcomes through other pathways than DNA methylation

**Table 1 Characteristics for the participating studies in the main meta-analysis for the association between neonatal blood DNA methylation and birthweight**

| Study | Total N | Normal birthweight, N (%) | High birthweight, N (%) | Birthweight (g) | Gestational age (wk) | Ethnicity | Boys, N (%) |
|---|---|---|---|---|---|---|---|
| ALSPAC | 633 | 547 (86.4) | 79 (12.5) | 3512 ± 443 | 39.7 ± 1.3 | European | 301 (47.6) |
| CBC[a]: Hispanic | 127 | 106 (83.5) | 19 (15.0) | 3445 ± 484 | 39.8 ± 1.3 | Hispanic | 74 (58.3) |
| CBC[a]: Caucasian | 136 | 108 (79.4) | 26 (19.1) | 3625 ± 472 | 39.7 ± 1.5 | European | 79 (58.1) |
| CHAMACOS | 283 | 236 (83.4) | 44 (15.5) | 3520 ± 446 | 39.3 ± 1.2 | Hispanic | 142 (50.1) |
| CHS[a] | 199 | 168 (84.4) | 28 (14.1) | 3486 ± 476 | 40.2 ± 1.2 | Mixed | 79 (39.7) |
| EARLI | 131 | 113 (86.3) | 16 (12.2) | 3507 ± 480 | 39.3 ± 1.0 | Mixed | 70 (53.4) |
| EXPOsOMICS: Rhea, Environage and Piccolipiu | 324 | 297 (91.7) | 22 (6.8) | 3368 ± 437 | 39.4 ± 1.2 | European | 169 (52.1) |
| GECKO | 255 | 206 (80.8) | 46 (18.0) | 3543 ± 533 | 39.7 ± 1.3 | European | 136 (53.3) |
| Gen3G | 162 | 145 (89.5) | 15 (9.3) | 3408 ± 431 | 39.5 ± 1.1 | European | 74 (45.7) |
| Generation R | 717 | 589 (82.1) | 122 (17.0) | 3572 ± 465 | 40.2 ± 1.1 | European | 372 (51.9) |
| GOYA[b] | 947 | 649 (68.5) | 294 (31.0) | 3750 ± 501 | 40.4 ± 1.3 | European | 483 (51.0) |
| Healthy Start: African American | 77 | – | – | 3059 ± 358 | 38.9 ± 1.3 | African American | 42 (54.5) |
| Healthy Start: Hispanic | 115 | – | – | 3322 ± 395 | 39.1 ± 1.1 | Hispanic | 55 (47.8) |
| Healthy Start: Caucasian | 240 | 220 (91.7) | 14 (5.8) | 3325 ± 425 | 39.3 ± 1.1 | European | 125 (52.1) |
| INMA | 166 | – | – | 3297 ± 400 | 39.9 ± 1.2 | European | 82 (49.4) |
| IOW F2 | 118 | 97 (82.2) | 17 (14.4) | 3432 ± 525 | 39.7 ± 1.6 | European | 59 (50.0) |
| MoBa1 | 1066 | 795 (74.6) | 251 (23.5) | 3644 ± 544 | 39.5 ± 1.6 | European | 568 (53.3) |
| MoBa2 | 587 | 435 (74.1) | 146 (24.9) | 3701 ± 487 | 40.1 ± 1.2 | European | 329 (56.0) |
| MoBa3 | 205 | 153 (74.6) | 51 (24.9) | 3706 ± 491 | 39.8 ± 1.2 | European | 106 (51.7) |
| NCL[a] | 792 | 592 (74.7) | 192 (24.2) | 3671 ± 506 | 40.0 ± 1.3 | European | 453 (57.2) |
| NEST: African American | 99 | – | – | 3197 ± 534 | 39.3 ± 1.2 | African American | 47 (47.5) |
| NEST: Caucasian | 111 | 94 (84.7) | 13 (11.7) | 3446 ± 471 | 39.5 ± 1.2 | European | 50 (45.0) |
| NHBCS | 96 | 84 (87.5) | 12 (12.5) | 3509 ± 453 | 39.6 ± 1.1 | European | 53 (55.2) |
| PREDO | 540 | 428 (79.3) | 99 (18.3) | 3572 ± 478 | 40.1 ± 1.2 | European | 264 (48.8) |
| PRISM | 138 | – | – | 3385 ± 441 | 39.5 ± 1.1 | Mixed | 76 (55.1) |
| PROGRESS | 143 | – | – | 3124 ± 387 | 38.6 ± 1.1 | Hispanic | 77 (53.8) |
| RICHS | 89 | 52 (58.4) | 23 (25.8) | 3335 ± 734 | 38.9 ± 1.2 | European | 35 (39.3) |
| Project Viva | 329 | 263 (79.9) | 64 (19.5) | 3623 ± 473 | 40.0 ± 1.2 | European | 168 (51.1) |
| Total N | 8825 | 6377 | 1593 | | | | |

Results are presented as mean ± SD or N (%). Normal birthweight: 2500−4000 g, high birthweight: >4000 g, low birthweight: <2500 g. Studies with mixed ethnicities analysed all participants together with adjustment for ethnicities. g: grams, wk: weeks, y: years. Full study names can be found in study-specific Supplementary Methods. For some studies the sample size for defining normal/high BW was too small
[a]CBC, CHS and NCL used heel prick blood spot samples instead of cord blood
[b]GOYA is a case-cohort study (cases are mothers with BMI>32 and controls are mothers randomly sampled from the underlying study population in which the cases were identified), in analyses where we included a random sample with a normal BMI distribution results were essentially the same as in the main analyses

the two African cohorts revealed that 94.9% and 74.0% of the 914 CpGs showed consistent direction of association, with Pearson correlation coefficients for point estimates of 0.82 and 0.48, respectively (Supplementary Data 3). In leave-one-out analyses, in which we reran the main meta-analysis repeatedly with one of the 24 studies removed each time, there was no strong evidence that any one study influenced findings consistently across the 914 differentially methylated CpGs that passed Bonferroni correction and for which between-study heterogeneity had an $I^2 \leq 50\%$. For 139/914 CpGs (15.2%) the difference in mean birthweight for a 10% greater methylation at that site varied by ≥20% with removal of a study, but the study resulting in the change was different for different CpGs. Supplementary Fig. 3.1-3.20 show the results for a random 10 plots where removal of one study changed the result by 20% or more and a random 10 where this was not the case; full results are available on request from the authors. Findings were broadly consistent when birthweight was categorised to high (>4000 g, $n = 1593$) versus normal (2500–4000 g, $n = 6377$) (Supplementary Data 4, yellow track (5) in Fig. 4) and when we did not exclude neonates born preterm or to women with pre-eclampsia or diabetes (Supplementary Fig. 4 and Supplementary Data 5A and 5C, and red track (3) in Fig. 4). Without these exclusions, we were able to examine associations with low (<2500 g, $n = 178$) versus normal (2500–4000 g, $n = 4197$) birthweight, though statistical power was

still limited. Four CpGs were associated with low versus normal birthweight (Bonferroni-corrected threshold), none of which over-lapped with the 914 CpGs from the main analysis (Supplementary Data 5B, purple track (4) in Fig. 4). We identified that 161 of the 914 differentially methylated CpGs potentially contained a single-nucleotide polymorphism (SNP) at cytosine or guanine positions (i.e. polymorphic CpGs; Supplementary Data 6). Polymorphic CpGs may affect probe binding and hence measured DNA methylation levels[26,27]. We used one of the largest studies (ALSPAC; $n = 633$) to explore this. We found no indication of bimodal distributions for any of the 161 CpGs suggesting SNPs had not markedly affected methylation measurements at these sites (dip test $p$-values: 0.299–1.00)[28–30].

**Analyses at older ages.** We took the 914 neonatal blood CpGs that were associated with birthweight at Bonferroni-corrected statistical significance and with $I^2 \leq 50\%$ and examined their associations with birthweight when measured in blood taken in childhood (2–13 years; $n = 2756$ from 10 studies), adolescence (16–18 years; $n = 2906$ from six studies) and adulthood (30–45 years; $n = 1616$ from three studies). Only participants from ALSPAC, CHAMACOS and Generation R had also contributed to the main neonatal blood EWAS. In childhood, adolescence and adulthood, we observed 87, 49 and 42 of the 914 CpGs to be nominally associated with birthweight ($p < 0.05$).

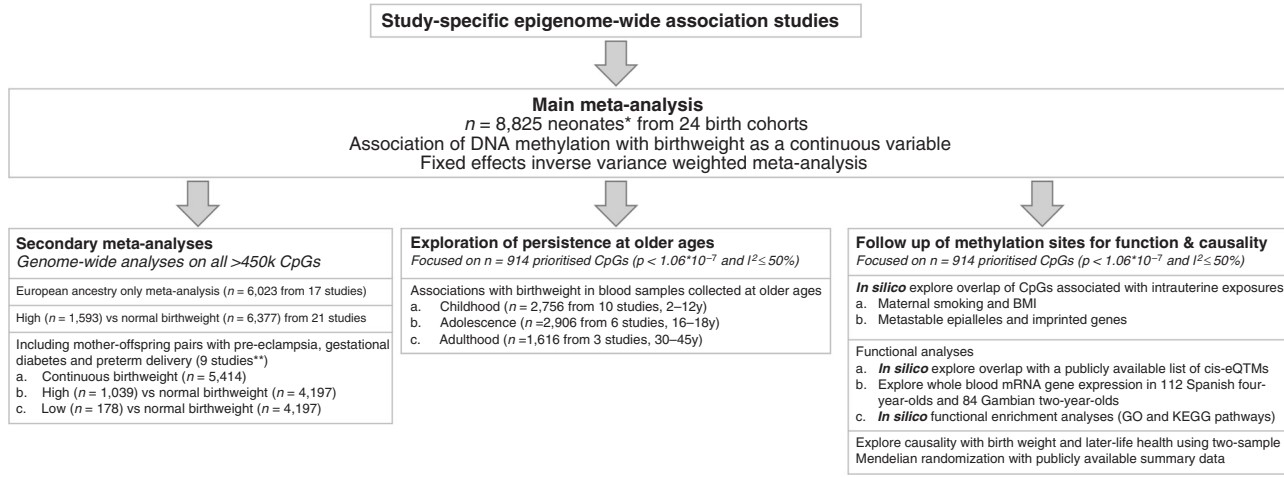

**Fig. 2** Design of the study. Schematic representation of the main meta-analysis, secondary meta-analyses, follow-up analyses and exploration of persistence at older ages. *We removed multiple births from all analyses and excluded preterm births (<37 weeks) and offspring of mothers with pre-eclampsia or diabetes (three major pathological causes of differences). **For sufficient power in the low vs normal BW analyses, we only included nine studies with >10 low birthweight cases

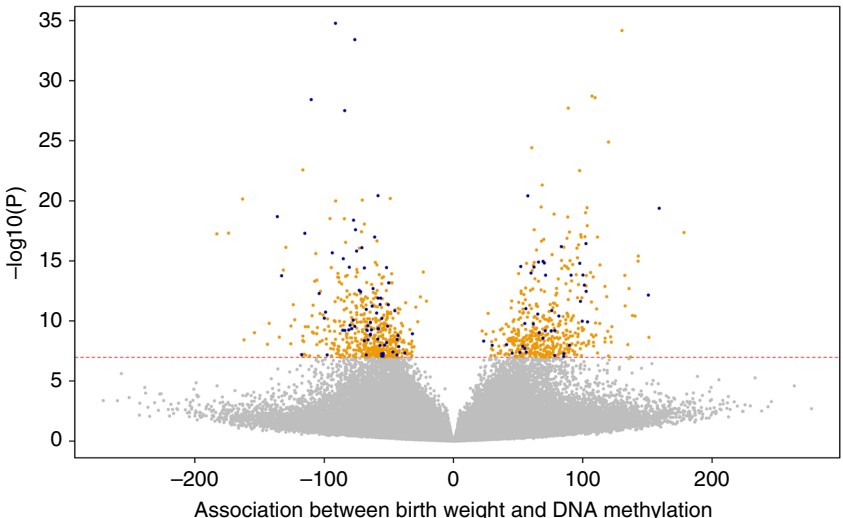

**Fig. 3** Volcano plot showing the direction of associations of DNA methylation with birthweight in 8825 neonates from 24 studies. The X-axis represents the difference in birthweight in grams per 10% methylation difference, the Y-axis represents the $-\log_{10}(P)$. The red line shows the Bonferroni-corrected significance threshold for multiple testing ($p < 1.06 \times 10^{-7}$). Highlighted in orange are the 914 CpGs with $p < 1.06 \times 10^{-7}$ and $I^2 \leq 50\%$ and highlighted in blue are the 115 CpGs with $p < 1.06 \times 10^{-7}$ and $I^2 > 50\%$

All these CpGs showed consistent directions of association. Ten CpGs showed differential methylation across all four age periods. However, only a minority survived Bonferroni correction for 914 tests ($p < 5.5 \times 10^{-5}$): 12 (1.3%), 1 (0.1%) and 0 CpGs in childhood, adolescence and adulthood, respectively (Supplementary Data 7; the 12 CpGs that persisted in childhood are presented in the green track (6) in Fig. 4). Of the 914 CpGs, 50, 52 and 49% showed consistency in direction of association in childhood, adolescence and adulthood, but correlations of the associations of DNA methylation and birthweight between methylation measured in infancy and that measured in childhood, adolescence and adulthood were weak (Pearson correlation coefficients: 0.15, 0.06 and 0.02, respectively).

**Intrauterine factors**. We observed enrichment of previously published maternal smoking-related CpGs in the birthweight-associated CpGs[14] (55/914 (6.0%) $p_{enrichment} = 6.12 \times 10^{-74}$, of which cg00253658 and cg26681628 also showed persistent methylation

differences in the look-up in childhood). We additionally found enrichment of maternal BMI-related CpGs in the list of birthweight-related CpGs[15] (3/914 (0.3%) $p_{enrichment} = 1.13 \times 10^{-3}$). All directions of association were consistent with the birthweight-lowering influence of maternal smoking or the positive association of maternal BMI with birthweight (Supplementary Data 8). We did not find evidence for overlap with plasma folate[31]. For famine, we were unable to explore overlap with DNA methylation at the Bonferroni-significant level as the previous EWAS of famine only reported results that reached a FDR level of statistical significance[32]. In additional analyses for overlap between all FDR hits from the birthweight EWAS with those FDR hits presented in the smoking, maternal BMI, folate and famine EWAS, we found an overlap of 430/8170 CpGs (5.3%, $p_{enrichment} = 7.38 \times 10^{-132}$) for smoking, 584/8170 CpGs (7.1%, $p_{enrichment} = 3.34 \times 10^{-62}$) for maternal BMI and 14/8170 (0.2%, $p_{enrichment} = 0.02$) for folate. For famine we did not observe overlap.

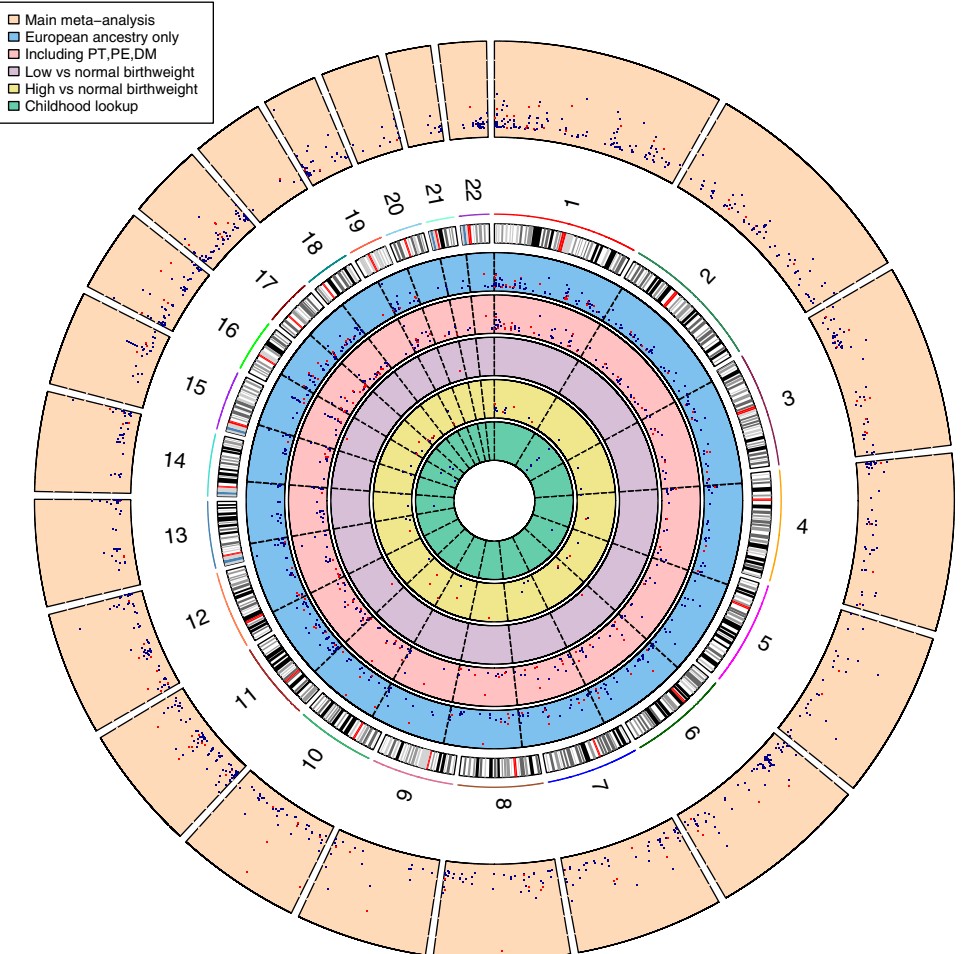

**Fig. 4** Circos plot showing the (Bonferroni-corrected $p < 1.06 \times 10^{-7}$) results for associations of DNA methylation with birthweight. Results are presented as CpG-specific associations ($-\log_{10}(P)$, each dot represents a CpG) by genomic position, per chromosome. From outer to inner track: [1, orange] Main analysis results for associations between DNA methylation and birthweight as a continuous measure ($n = 8825$), [2, blue] Results from participants from European ethnicity only, DNA methylation and birthweight as a continuous measure ($n = 6023$), [3, red] Results from analysis without exclusion for preterm births, pre-eclampsia and maternal diabetes, DNA methylation and birthweight as a continuous measure $n = 5414$), [4, purple] Results from logistic regression analysis without exclusion for preterm births, pre-eclampsia and maternal diabetes, for low ($n = 178$) vs normal ($n = 4197$) birthweight, [5, yellow] Results from logistic regression analysis for associations between DNA methylation and high ($n = 1590$) vs normal ($n = 6114$) birthweight, [6, green] Results from look-up analysis in methylation samples taken during childhood and its association with birthweight as a continuous measure ($n = 2756$). Track 1: highlighted in red are 115 CpGs with $I^2 > 50\%$. Tracks 2–6: highlighted in red are CpGs that were not found in the 914 main meta-analysis hits (though note differences in sample size and hence statistical power for different analyses presented in the different tracks)

**Metastable epialleles and imprinted genes**. We tested the birthweight-associated CpGs for enrichment of metastable epialleles (loci for which the methylation state is established in the periconceptional period[33,34]). We additionally tested for enrichment of CpGs annotated to imprinted genes (loci that depend on the maintenance of parental-origin-specific methylation marks in the pre-implantation embryo, some of which are known to regulate fetal growth[35,36]). We did not find evidence of enrichment for metastable epialleles (3/1936 metastable epialleles overlap a birthweight-associated CpG), imprinting control regions (0/741) or imprinted gene transcription start sites (5/1728) (Supplementary Data 9).

**Comparison with GWAS for birthweight**. To compare these EWAS results to those from genetic studies, we used the 60 recently published fetal SNPs associated with birthweight in a GWAS meta-analysis of 153,781 newborns[37] and mapped the CpG sites identified in the EWAS to these SNPs to seek evidence of co-localisation of

genetic and epigenetic variation (Supplementary Data 10). We repeated this for the 10 recently published maternal SNPs associated with birthweight in a GWAS meta-analysis of 86,577 women[38] (Supplementary Data 11). We observed that one or more of the 914 birthweight-associated CpGs were within +/−2Mb of 34/60 fetal and all 10 maternal birthweight-associated SNPs. Of the 34 fetal SNPs, three were located in the same gene as the CpG, as was one of the ten maternal SNPs. Ten fetal and four maternal SNPs were within 100 kb of identified CpGs. In a look-up of the fetal and maternal SNPs from GWAS of birthweight in an online cord blood methylation quantitative trait loci (mQTL) database (mqtldb.org[39]), 35 fetal and four maternal SNPs affected methylation at some CpG(s), but none at the 914 birthweight-associated CpGs specifically.

**Functional analyses**. We compared the 914 birthweight-related CpGs with a recently published list of 18,881 expression quantitative trait methylation sites (cis-eQTMs, +/−250 kb around

the transcription start site), CpG sites known to correlate with gene expression, from whole blood samples of 2101 Dutch adult individuals. We found that 82 of the 914 birthweight-associated CpGs were associated with gene expression of 98 probes (cis-eQTMs)[40] ($p_{enrichment} < 1.73 \times 10^{-11}$, Supplementary Data 12). Additionally, in 112 Spanish 4-year-olds[41], we observed that 19 CpGs were inversely associated with whole blood mRNA gene expression and four CpGs were positively associated with gene expression (FDR<0.05, Supplementary Data 13). Of these 23 CpGs, 13 were also found in the publicly available cis-eQTM list[40]. In 84 Gambian children (age 2 years)[42], we found two CpGs that were inversely associated with whole blood mRNA gene expression, but neither were found in the Spanish results or the publicly available cis-eQTM list. The 914 birthweight-associated CpGs showed no functional enrichment of Gene Ontology (GO) terms or Kyoto Encyclopedia of Genes and Genomes (KEGG) terms (FDR<0.05).

**Mendelian randomization**. We aimed to explore causality using MR analysis, in which genetic variants associated with methylation levels (methylation quantitative trait loci (mQTLs)) are used as instrumental variables to appraise causality. For 788 (86%) of the 914 birthweight-associated CpGs, no mQTLs were identified in a publicly available mQTL database[39]. For 108 (86%) of the remaining 126 CpGs, only one mQTL was identified and for the remainder none had more than four mQTLs (Supplementary Data 14 provides a complete list of all mQTLs identified for these 126 CpGs). Many of the currently available methods that can be used as sensitivity analyses to explore whether MR results are biased by horizontal pleiotropy (a single mQTL influencing multiple traits) require more than one genetic instrument (here mQTLs) and even with two or three this can be difficult to interpret[43]. Having determined that it was not possible to undertake MR analyses of 86% of the birthweight-related differentially methylated CpGs (because we did not identify any mQTLs), and for the majority of the remaining CpGs we would not have been reliably able to distinguish causality from horizontal pleiotropy (because only one mQTL could be identified), we decided not to pursue MR analyses further.

## Discussion

This large-scale meta-analysis shows that birthweight is associated with widespread differences in DNA methylation. We observed some enrichment of birthweight-associated CpGs among sites that have previously been linked to smoking during pregnancy[14] and pre-pregnancy BMI[15], consistent with the hypothesis that epigenetic pathways may underlie the observational associations of those prenatal exposures with birthweight[21,44,45]. However, the actual overlap in this analysis was modest, likely explained by the adjustments for maternal smoking and BMI in the EWAS analyses. The overlap that we observed with pregnancy smoking-related CpGs may reflect the possibility that smoking-related CpGs capture smoking better than self-report[46,47], in line with expectations of pregnant women underreporting their smoking behaviour. Adjustment for maternal smoking and BMI may have masked a greater level of overlap between our results and EWAS of these two maternal exposures. The fact that we find an association of DNA methylation across the genome with birthweight provides some support for our conceptual framework shown in Fig. 1. However, we acknowledge that the associations that we have observed may also be explained by causal effects of maternal pregnancy exposures on both DNA methylation and fetal growth, as well as subtle inflammatory responses in cell-type proportions associated with maternal smoking that might not have been completely captured with the currently available cell type estimation methods.

The differential methylation associated with birthweight in neonates persisted only minimally across childhood and into adulthood.

Larger (preferably longitudinal) studies are needed to explore persistent differential methylation in more detail and with better power at older ages. It is possible that inclusion of the Gambia study in the childhood EWAS (which was the only non-European study in these analyses and was not included in the main meta-analyses with neonatal blood) might have impacted these results, although this study made up just 7% of the total child follow-up sample. A rapid attenuation of differential methylation in relation to birthweight in the first years after birth has previously been reported[19], but our sample size for these analyses may have been too small to detect persistence. This rapid decrease, if real, may indicate a reduction in the dose of the child's exposure to maternal factors such as smoking once the offspring is delivered, with that reduction continuing as the child ages. Persistence of birthweight-related differential DNA methylation may not necessarily be a prerequisite for long-term effects, as transient differential methylation in early life may cause lasting functional alterations in organ structure and function that predispose to later adverse health effects.

Methylation is known to be associated with gene expression[48]. However, we found no consistent associations between birthweight-related methylation and gene expression in two childhood studies. This could be due to the relatively small sample sizes, differences in ethnicities, age, or platforms to measure gene expression. The use of blood, which is likely only a possible surrogate tissue for fetal growth phenotypes, for gene expression analysis might also explain the lack of findings. We did find multiple cis-eQTMs among the birthweight-related CpGs at which methylation was related to gene expression in blood when using a publicly available database from a larger adult sample[40], providing some evidence that birthweight-related differentially methylated CpGs may be associated with gene expression. These initial in silico association analyses need further exploration to establish any underlying causal mechanisms.

In observational studies, birthweight has repeatedly been associated with a range of later-life diseases. Change in DNA methylation has been hypothesized as a potential mechanism linking early exposures, birthweight and later health (Fig. 1). We originally aimed to explore this using MR analysis. For the vast majority of the birthweight-associated CpGs, no genetic instrumental variables were available. For the remaining 126 CpGs, only one mQTL was available, which would make it impossible to disentangle causality from horizontal pleiotropy. To ensure a strong basis for future MR analyses on this topic, there is a clear need for a more extensive mQTL resource.

Strengths of this study are its large sample size and the extensive analyses that we have undertaken. In a post hoc power calculation based on the sample size of 8825 with a weighted mean birthweight of 3560 g (weighted mean standard deviation (SD): 483 g) and with an alpha set at the Bonferroni-corrected level of $P < 1.06 \times 10^{-7}$ we had 80% power, with a two-sided test, to detect a minimum difference of 0.13 SD (63 g) in birthweight for each SD increase in methylation. The difference in methylation corresponding to a 1 SD increase differs per CpG, as it depends on the distribution of the methylation values. We acknowledge that smaller differences which might be clinically or biologically relevant may not have been identified in the current analysis. Nonetheless, to our knowledge this analysis has brought together all studies currently available with relevant data and is the largest published study of this association. DNA methylation patterns in neonatal blood, whilst easily accessible in large numbers, may not reflect the key tissue of importance in relation to birthweight. DNA methylation and gene expression in placental tissue may be important targets for future studies. DNA methylation varies between leucocyte subtypes[49] and we used an adult whole blood reference to correct for this in the main analyses[23,24], as the study-specific analyses were completed before the widespread availability of specific cord blood reference datasets[50,51]. However, we observed very similar findings in two studies

(Generation R and GECKO) when we compared the results with those using one of the currently available cord blood references[50]. Although we adjusted for potential major confounders that may affect both methylation and fetal growth, we acknowledge that the main results cannot ascertain causality. That is, whilst we have hypothesised that variation in fetal DNA methylation influences fetal growth and hence birthweight, and undertaken the analyses accordingly, we cannot exclude the possibility that differences in neonatal blood DNA methylation are caused by variation in fetal growth itself, or that the association is confounded by factors, including maternal smoking and BMI, that independently influence both fetal growth and DNA methylation (as suggested in Fig. 1). The 450k array that was used to measure genome-wide DNA methylation only covers 1.7% of the total number of CpGs present in the genome and specifically targets CpGs in promoter regions and gene bodies[52]. We removed the CpGs that were flagged as potentially cross-reactive, as the measured methylation levels may represent methylation at either of the potential loci. Also, although we did not find evidence for polymorphic effects for the 161 potentially polymorphic CpGs in ALSPAC, we cannot completely exclude these potential polymorphic effects in the meta-analysed results. The majority of participants were of European ethnicity and when analyses were restricted to those of European ethnicity the results were essentially identical to those with all studies included. Direct comparisons of the main analysis with analyses in those of Hispanic or of African ethnicity for the 914 hits suggested strong correlations with Hispanic but weaker with African ethnicity. However, these results need to be treated with caution. First, we had very few studies of Hispanic and African populations. Second, we only compared the initial hits from the main meta-analysis with all ethnicities included. A detailed exploration of ethnic differences would require similar large samples for each ethnic group and within ethnic EWAS, which is beyond the scope of the data currently available.

Neonatal blood DNA methylation at many sites across the genome is associated with birthweight. Further research is required to determine if these are causal and if so whether they mediate any long-term effect of intrauterine exposures on future health.

## Methods

**Participants**. In the main EWAS meta-analysis we explored associations of neonatal blood DNA methylation with birthweight using data from 8825 neonates from 24 studies in the PACE Consortium[53] (Table 1). We removed multiple births from all analyses and excluded preterm births (<37 weeks) and offspring of mothers with pre-eclampsia or diabetes (three major pathological causes of differences in fetal growth). In follow-up analyses, we explored whether any sites found in the main analysis were discernible in relation to birthweight when examined in DNA from blood drawn during childhood (2–13 years; 2756 children from 10 studies), adolescence (16–18 years; 2906 adolescents from six studies) or adulthood (30–45 years; 1616 adults from three studies), see Supplementary Data 1B. Informed consent was obtained from all participants, and all studies received approval from local ethics committees. Study-specific methods and ethical approval statements are provided in Supplementary Methods.

**Birthweight, DNA methylation and covariates**. Our primary outcome was birthweight on a continuous scale (grams), adjusted for gestational age, and measured immediately after birth or retrospectively reported by mothers in questionnaires. In secondary analyses, we categorised and compared associations with high (>4000 g, $n = 1593$) versus normal (2500–4000g, $n = 6377$) birthweight. We also explored all associations with (continuous and categorical) birthweight in analyses that did not exclude women with pre-eclampsia, diabetes or preterm delivery, which also resulted in enough cases to explore low (<2500 g, $n = 178$) versus normal (2500–4000 g, $n = 4197$) birthweight (Supplementary Data 1C shows the characteristics of participants). Primary, secondary and follow-up analyses are outlined in the study design in Fig. 2. DNA methylation was measured in neonatal blood samples using the Illumina Infinium® HumanMethylation450 BeadChip assay. All participants had cord blood samples except for three studies with heel stick blood spots ($n = 1254$ [14.2%]). After study-specific laboratory analyses, quality control, normalisation, and removal of control probes ($n = 65$) and probes that mapped to the X ($n = 11,232$) and Y ($n = 370$) chromosomes, we included 473,864 CpGs. DNA methylation is expressed as the proportion of cells in which the DNA was methylated at a specific site and hence takes values from zero to one. We converted this to a percentage and present differences in

mean birthweight per 10% higher DNA methylation level at each CpG. All analyses were adjusted for gestational age at delivery, child sex, maternal age at delivery, parity (0/≥1), smoking during pregnancy (no smoking/stopped in early pregnancy/smoking throughout pregnancy), pre-pregnancy BMI, socio-economic position, technical variation, and estimated white blood cell proportions (B-cells, CD8+ T-cells, CD4+ T-cells, granulocytes, NK-cells and monocytes)[23–25]. In studies with participants from multiple ethnic groups, each group was analysed separately and results were added to the meta-analyses as separate studies. Further details are provided in the study-specific Supplementary Methods.

**Statistical methods**. Robust linear (birthweight as a continuous outcome) or logit (binary birthweight outcomes) regression EWAS were undertaken within each study according to a pre-specified analysis plan. Quality control, normalisation and regression analyses were conducted independently by each study. After confirming comparability of study-specific summary statistics[54], we combined results using a fixed effects inverse variance weighted meta-analysis[55]. The meta-analysis was done independently by two study groups and the results were compared in order to minimise the likelihood of human error. We show (two-sided) results after correcting for multiple testing using both the FDR<0.05[56] and the Bonferroni correction ($p < 1.06 \times 10^{-7}$). We completed follow-up analyses for differentially methylated CpGs that reached the Bonferroni-adjusted threshold and did not show large between-study heterogeneity[57] ($I^2 \le 50\%$). We annotated the nearest gene for each CpG using the UCSC Genome Browser build hg19[58,59]. We explored whether between-study heterogeneity might be explained by differences in ethnicity between studies, by repeating the meta-analysis including only participants of European ethnicity, which was by far the largest ethnic subgroup ($n = 6023$ from 17 studies) (Fig. 2). Ethnicity was defined using maternal or self-report, unless specified otherwise in study-specific Supplementary Methods. We also did meta-analyses only including the Hispanic studies and only including the African American studies and present those results for illustrative purposes only, given the much smaller sample size. All analyses were performed using R[60], except for the meta-analysis which was performed using METAL[55]. We removed CpGs that co-hybridised to alternate sequences (i.e. cross-reactive sites), because we cannot distinguish whether the differential methylation is at the locus that we have reported or at the one that the probe cross-reacts with. We compared the birthweight-related CpGs to lists of CpGs that are potentially influenced by a SNP (polymorphic sites)[26,27]. For these CpGs, we determined if DNA methylation levels were influenced by nearby SNPs, by assessing whether their distributions deviated from unimodality using Hartigans' dip test[28,29] and visual inspection of density plots in $n = 742$ cord blood samples in the ALSPAC study.

**Analyses at older ages**. Analyses of the associations with DNA methylation in blood collected in childhood, adolescence and adulthood followed the same covariable adjustment and methods as for the main analyses ($p < 5.5 \times 10^{-5}$ for 914 tests). All participants and studies in these analyses at older ages had not been included in the main meta-analysis in neonatal blood, except for ALSPAC ($n = 633$ in neonatal analyses, $n = 605$ in childhood and $n = 526$ in adolescence), CHAMACOS ($n = 283$ in neonatal analyses and $n = 191$ in childhood) and Generation R ($n = 717$ in neonatal analyses and $n = 372$ in childhood). Characteristics are shown in study-specific Supplementary Methods and Supplementary Data 1B.

**Intrauterine factors**. We used a hypergeometric test to explore the extent to which the birthweight-related CpGs overlapped with those previously associated with intrauterine exposure to smoking[14] ($n = 568$ CpGs), BMI[15] ($n = 104$ CpGs) and plasma folate[31] ($n = 48$ CpGs), using the same (Bonferroni-corrected) cut-off for statistical significance. No CpGs reached the Bonferroni-corrected cut-off for famine[32]. We additionally appraised this overlap using the FDR<0.05 cut-off for all traits ($n = 8170$ birthweight-related CpGs, $n = 6703$ smoking-related CpGs, $n = 16,067$ BMI-related CpGs, $n = 443$ folate-related CpGs, $n = 7$ famine-related CpGs). These FDR results were available from the publications for smoking, folate and famine, and we obtained them from the corresponding author for BMI.

**Metastable epialleles and imprinted genes**. We tested the birthweight-associated CpGs for enrichment of metastable epialleles and CpGs associated with imprinted genes. The metastable epialleles were derived from a recently published study that identified 1936 putative metastable epialleles[34]. For imprinted genes, we first identified a set of CpGs falling within a curated set of imprinting control regions; differentially methylated regions controlling the parental-specific expression of one or more imprinted genes[36]. Second, we extracted the set of imprinting control region controlled genes from the above source and identified all 450k CpGs within +/−10kbp of the gene transcription start site, including all known alternative TSS identified in grch37.ensembl.org using biomaRt[61,62].

**Comparison with GWAS for birthweight**. We compared the birthweight-associated CpGs with the 60 SNPs from the most recent GWAS meta-analyses of fetal genotype associations with birthweight in >150,000 newborns[37] and with the 10 SNPs from the most recent GWAS meta-analysis of maternal genotype associations with birthweight in >86,000 women[38]. With this comparison we checked if the EWAS top hits were located within a 4 Mb window (+/− 2 Mb)

surrounding these SNPs. We additionally checked whether SNPs and CpGs were located in the same gene.

**Functional analyses.** To explore the association of methylation with gene expression, we compared birthweight-related CpGs with a recently published list of 18,881 cis-eQTMs from whole blood samples of 2101 Dutch adult individuals[40]. With a hypergeometric test, we calculated enrichment of cis-eQTMs in the list of birthweight-associated CpGs. We further explored methylation of birthweight-associated CpGs in relation to whole blood mRNA gene expression (transcript levels) within a 500 kb region of the CpGs (+/−250 kb, FDR<0.05) in 112 Spanish 4-year-olds[41] and 84 Gambian 2-year-olds[42] (Supplementary Methods). To better understand the potential mechanisms linking DNA methylation and birthweight, we explored the potential functions of the birthweight-associated CpGs using GO and Kyoto Encyclopedia of Genes and Genomes (KEGG) enrichment analyses. We used the missMethyl R package[63], which enabled us to correct for the number of probes per gene on the 450k array, based on the November 2018 version of the GO and KEGG source databases. To filter out the large, general pathways we set the number of genes for each gene set between 15 and 1000, respectively. We calculated FDR at 5% corrected P-values for enrichment.

**Mendelian randomization.** MR uses genetic variants as instrumental variables to study the causal effect of exposures on outcomes[64,65]. We aimed to use two-sample MR[22,66] to explore (a) evidence of a causal association of methylation levels at the identified CpGs with birthweight and (b) evidence of a causal association of these CpGs with later-life health outcomes (i.e. to explore our hypothesised causal mechanisms shown in Fig. 1). We did this by first searching a publicly available mQTL database[39] to identify cis-mQTLs within 1 Mb of each of the Bonferroni-corrected, with $I^2 \leq 50\%$, birthweight-related differentially methylated CpGs. These mQTLs could then be used as genetic instrumental variables for methylation levels of the birthweight-related CpGs. We then aimed to determine the association of these mQTLs with birthweight and later-life health outcomes from publicly available summary GWAS results[66].

**Reporting summary.** Further information on experimental design is available in the Nature Research Reporting Summary linked to this article.

## Data availability

All relevant data supporting the key findings of this study are available within the article and its Supplementary Information files or from the corresponding authors upon reasonable request. All summary statistics from this EWAS meta-analysis are available via doi: 10.5281/zenodo.2222287. A reporting summary for this Article is available as a Supplementary Information file.

## Code availability

The code used for this EWAS meta-analysis is available from the authors upon request.

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

## Acknowledgements

For all studies, acknowledgements can be found in Supplementary Information: Supplementary Acknowledgements. For all studies, funding statements can be found in Supplementary Information: Supplementary Funding.

## Author contributions

L.K.K., D.A.L., C.L.R., H.S. and J.F.F. conceived and designed the study. Study-specific analyses were completed by G.C.S. (ALSPAC and GOYA), S.K.M. (BAMSE), R.R. (CBC), P.Y. (CHAMACOS), C.V.B. (CHS), K.M.B. (EARLI), A.G. and A.N. (EXPOsOMICS, The Gambia and MoBa3), S.A.S.L. (FLEHS1), L.K.K. (GECKO), C.A. (Gen3G), C.M. (Generation R), J.L. (Glaku), A.P.S. (Healthy Start), L.A.S. (INMA), F.I.R. (IOW F1), J.W.H. (IOW F2), D.A.V.D. P. (Lifelines), C.M.P. (MoBa1), S.E.R. (MoBa2), A.J.W. (NCL), D.D.J. (NEST), M.W. (NFBC66 and NFBC86), T.M.E. (NHBCS and RICHS), J.V.D. (NTR), C.J.X. (PIAMA), D.C. (PREDO), A.C.J. (PRISM), A.C.J. (PROGRESS), S.L. (Project Viva), R.C.H. (Raine), V.U. (STOPPA). L.K.K. and C.M. meta-analysed the results. L.K.K., C.M., G.C.S., P.Y., L.A.S., A.G., A.N. and M.J.S. performed follow-up analyses. L.K.K., D.A.L., C.L.R., H.S. and J.F.F. interpreted the results. L.K.K., with input from D.A.L., C.L.R., H.S. and J.F.F., wrote the first draft of the manuscript. All authors (L.K.K., C.M., G.C.S., P.Y., L.A.S., A.G., C.M.P., S.E.R., A.J.W., D.C., A.P.S., A.N., S.L., R.R., C.H., C.V.B., C.A., A.C.J., K.M.B., J.W.H., T.M.E., C.-J.X., R.-C.H., D.Avd.P., M.W., S.K.M., V.U., F.I.R., J.L., J.v.D., S.A.S.L., T.G.R., M.C.M., E.A.N., Z.X., L.D., S.Z., W.Z., M.P., D.L.D., O.S., J.H.H., D.D.J., L.G., M.B., P.P., R.O.W., I.H-P., H.Z., M.R.K., U.G., C.J.M., L.J.B., J.M.V., M.-R.J., A.B., A.K.Ö., S.E., P.M.V., S.E.M., G.W., A.R.L.S., S.E.H., T.I.A.S., J.A.T., K.R., I.V.Y., K.K., T.S.N., M.J.S., Y.Y.G., L.R., M.K., A.A.L., B.E., K.H., H.M., R.L.M., T.D., M.V., L.B., A.A.B., L.A.C., W.K., D.A., M.dV., S.S., J.K., R.K., S.H.A., E.H., M.N.R., D.I.B., A.P.F., C.J.N., E.G., M.M., M.D.F., E.M., A.M.P., E.K., C.A., E.O., D.D., H.M.B., P.E.M., R.J.W., G.H.K., L.T., M.-F.H., J.S., M.C.M.-K., S.K.M., E.C., J.W., N.H., Z.H., E.B.B., G.D.S., V.W.V.J., R.T.L., W.N., S.J.L., D.A.L., C.L.R., H.S., J.F.F.) read and critically revised subsequent drafts.

## Additional information

**Competing interests:** The authors declare no competing interests.

Leanne K. Küpers[1,2,3,4], Claire Monnereau[5,6,7], Gemma C. Sharp[1,8], Paul Yousefi[1,2,9], Lucas A. Salas[10,11], Akram Ghantous[12], Christian M. Page[13,14], Sarah E. Reese[15], Allen J. Wilcox[15], Darina Czamara[16], Anne P. Starling[17], Alexei Novoloaca[12], Samantha Lent[18], Ritu Roy[19,20], Cathrine Hoyo[21,22], Carrie V. Breton[23], Catherine Allard[24], Allan C. Just[25], Kelly M. Bakulski[26], John W. Holloway[27,28], Todd M. Everson[29], Cheng-Jian Xu[30,31], Rae-Chi Huang[32], Diana A. van der Plaat[33], Matthias Wielscher[34], Simon Kebede Merid[35], Vilhelmina Ullemar[36], Faisal I. Rezwan[28], Jari Lahti[37,38], Jenny van Dongen[39], Sabine A.S. Langie[40,41,42], Tom G. Richardson[1,2], Maria C. Magnus[1,2,13], Ellen A. Nohr[43], Zongli Xu[44], Liesbeth Duijts[4,44,45], Shanshan Zhao[46], Weiming Zhang[47], Michelle Plusquin[48,49], Dawn L. DeMeo[50], Olivia Solomon[8], Joosje H. Heimovaara[3], Dereje D. Jima[22,51], Lu Gao[23], Mariona Bustamante[11,52,53,54], Patrice Perron[24,55], Robert O. Wright[25], Irva Hertz-Picciotto[56], Hongmei Zhang[57], Margaret R. Karagas[10,58], Ulrike Gehring[59], Carmen J. Marsit[29], Lawrence J. Beilin[60], Judith M. Vonk[33], Marjo-Riitta Jarvelin[34,61,62,63], Anna Bergström[35,64], Anne K. Örtqvist[36], Susan Ewart[65], Pia M. Villa[66], Sophie E. Moore[67,68], Gonneke Willemsen[39], Arnout R.L. Standaert[40], Siri E. Håberg[13], Thorkild I.A. Sørensen[1,69,70], Jack A. Taylor[15], Katri Räikkönen[38], Ivana V. Yang[71], Katerina Kechris[45], Tim S. Nawrot[48,72], Matt J. Silver[67], Yun Yun Gong[73], Lorenzo Richiardi[74,75], Manolis Kogevinas[11,53,54,76], Augusto A. Litonjua[50], Brenda Eskenazi[9,77], Karen Huen[9], Hamdi Mbarek[78], Rachel L. Maguire[21,79], Terence Dwyer[80], Martine Vrijheid[11,53,54], Luigi Bouchard[81,82], Andrea A. Baccarelli[83,84], Lisa A. Croen[85], Wilfried Karmaus[57], Denise Anderson[32], Maaike de Vries[33], Sylvain Sebert[61,62,86], Juha Kere[87,88,89], Robert Karlsson[36], Syed Hasan Arshad[27,90], Esa Hämäläinen[91], Michael N. Routledge[92], Dorret I. Boomsma[39,93], Andrew P. Feinberg[94], Craig J. Newschaffer[95], Eva Govarts[40], Matthieu Moisse[96,97], M. Daniele Fallin[98], Erik Melén[35,99], Andrew M. Prentice[67], Eero Kajantie[100,101,102], Catarina Almqvist[36,103], Emily Oken[104], Dana Dabelea[105], H. Marike Boezen[33], Phillip E. Melton[106,107], Rosalind J. Wright[25], Gerard H. Koppelman[30], Letizia Trevisi[108], Marie-France Hivert[55,104,109], Jordi Sunyer[11,53,54,76], Monica C. Munthe-Kaas[110,111], Susan K. Murphy[112], Eva Corpeleijn[3], Joseph Wiemels[113], Nina Holland[9], Zdenko Herceg[12], Elisabeth B. Binder[16,114], George Davey Smith[1,2], Vincent W.V. Jaddoe[5,6,7], Rolv T. Lie[13,115], Wenche Nystad[116], Stephanie J. London[15], Debbie A. Lawlor[1,2], Caroline L. Relton[1,2], Harold Snieder[3] & Janine F. Felix[5,6,7]

[1]MRC Integrative Epidemiology Unit, University of Bristol, Bristol, UK. [2]Population Health Sciences, Bristol Medical School, University of Bristol, Bristol, UK. [3]University of Groningen, University Medical Center Groningen, Department of Epidemiology, Groningen, The Netherlands. [4]Division of Human Nutrition and Health, Wageningen University, Wageningen, The Netherlands. [5]The Generation R Study Group, Erasmus MC, University Medical Center Rotterdam, Rotterdam, The Netherlands. [6]Department of Epidemiology, Erasmus MC, University Medical Center Rotterdam, Rotterdam, The Netherlands. [7]Department of Pediatrics, Erasmus MC, University Medical Center Rotterdam, Rotterdam, The Netherlands. [8]School of Oral and Dental Sciences, University of Bristol, Bristol, UK. [9]Children's Environmental Health Laboratory, Division of Environmental Health Sciences, School of Public Health, University of California, Berkeley, CA, USA. [10]Department of Epidemiology, Geisel School of Medicine at Dartmouth College, Hanover, NH, USA. [11]ISGlobal, Bacelona Institute for Global Health, Barcelona, Spain. [12]Epigenetics Group, International Agency for Research on Cancer, Lyon, France. [13]Centre for Fertility and Health, Norwegian Institute of Public Health, Oslo, Norway. [14]Oslo Centre for Biostatisitcs and Epidemiology, Oslo University Hospital, Oslo, Norway. [15]Epidemiology Branch, National Institute of Environmental Health Sciences, National Institutes of Health, Department of Health and Human Service, Research Triangle Park, Durham, NC, USA. [16]Department of Translational Research in Psychiatry, Max-Planck-Institute of Psychiatry, Munich, Germany. [17]Department of Epidemiology, Colorado School of Public Health, University of Colorado Anschutz Medical Campus, Aurora, CO, USA. [18]Department of Biostatistics, Boston University School of Public Health, Boston, MA, USA. [19]HDF Comprehensive Cancer Center, University of California, San Francisco, CA, USA. [20]Computational Biology and Informatics, UCSF, San Francisco, CA, USA. [21]Department of Biological Sciences, North Carolina State University, Raleigh, NC, USA. [22]Center for Human Health and the Environment, North Carolina State University, Raleigh, NC, USA. [23]Department of Preventive Medicine, University of Southern California, Los Angeles, CA 90089, USA. [24]Centre de recherche du Centre hospitalier universitaire de Sherbrooke, Sherbrooke, QC, Canada. [25]Department of Environmental Medicine and Public Health, Icahn School of Medicine at Mount Sinai, New York, NY, USA. [26]Department of Epidemiology, School of Public Health, University of Michigan, Ann Arbor, MI, USA. [27]Clinical and Experimental Sciences, Faculty of Medicine, University of Southampton, Southampton, UK. [28]Human Development and Health, Faculty of Medicine, University of Southampton, Southampton General Hospital, Southampton, UK. [29]Department of Environmental Health, Rollins School of Public Health at Emory University, Atlanta, GA, USA. [30]University of Groningen, University Medical Center Groningen, Department of Pediatric Pulmonology and Pediatric Allergology, Beatrix Children's Hospital, Groningen Research Institute for Asthma and COPD, Groningen, The Netherlands. [31]University of Groningen, University Medical Center Groningen, Department of Genetics, Groningen, The Netherlands. [32]Telethon Kids Institute, University of Western Australia, Perth, Australia. [33]University of Groningen, University Medical Center Groningen, Department of Epidemiology and Groningen Research Institute for Asthma and

COPD (GRIAC), Groningen, The Netherlands. [34]Department of Epidemiology and Biostatistics, MRC–PHE Centre for Environment & Health, School of Public Health, Imperial College London, London, UK. [35]Institute of Environmental Medicine, Karolinska Institutet, Stockholm, Sweden. [36]Department of Medical Epidemiology and Biostatistics, Karolinska Institutet, Stockholm, Sweden. [37]Helsinki Collegium for Advanced Studies, University of Helsinki, Helsinki, Finland. [38]Department of Psychology and Logopedics, Faculty of Medicine, University of Helsinki, Helsinki, Finland. [39]Department of Biological Psychology, Netherlands Twin Register, Vrije Universiteit Amsterdam, Amsterdam, The Netherlands. [40]VITO - Health, Mol, Belgium. [41]Theoretical Physics, Faculty of Sciences, Hasselt University, Hasselt, Belgium. [42]Centre for Environmental Sciences, Hasselt University, Hasselt, Belgium. [43]Research Unit for Gynaecology and Obstetrics, Department of Clinical Research, University of Southern Denmark, Odense, Denmark. [44]Department of Pediatrics, Division of Respiratory Medicine and Allergology, Erasmus MC, University Medical Center Rotterdam, Rotterdam, The Netherlands. [45]Department of Pediatrics, Division of Neonatology, Erasmus MC, University Medical Center Rotterdam, Rotterdam, The Netherlands. [46]Biostatistics and Computational Biology Branch, National Institute of Environmental Health Sciences, NIH, Research Triangle Park, Durham, NC, USA. [47]Department of Biostatistics and Informatics, Colorado School of Public Health, University of Colorado Anschutz Medical Campus, Aurora, CO, USA. [48]Centre for Environmental Sciences, Hasselt University, Diepenbeek, Belgium. [49]MRC/PHE Centre for Environment and Health School of Public Health Imperial College London, St Mary's Campus, Norfolk Place, London, UK. [50]Channing Division of Network Medicine, Brigham and Women's Hospital, Harvard Medical School, Boston, MA, USA. [51]Bioinformatics Research Center, North Carolina State University, Raleigh, NC, USA. [52]Bioinformatics and Genomics Program, Centre for Genomic Regulation (CRG), Barcelona, Spain. [53]Universitat Pompeu Fabra (UPF), Barcelona, Spain. [54]CIBER Epidemiología y Salud Pública (CIBERESP), Barcelona, Spain. [55]Department of Medicine, Universite de Sherbrooke, Sherbrooke, QC, Canada. [56]Department of Public Health Sciences, School of Medicine, University of California Davis MIND Institute, Sacramento, CA, USA. [57]Division of Epidemiology, Biostatistics and Environmental Health, School of Public Health, University of Memphis, Memphis, TN, USA. [58]Children's Environmental Health & Disease Prevention Research Center at Dartmouth, Hanover, NH, USA. [59]Institute for Risk Assessment Sciences, Utrecht University, Utrecht, The Netherlands. [60]Medical School, University of Western Australia, Perth, Australia. [61]Center for Life Course Health Research, Faculty of Medicine, University of Oulu, 90014 Oulu, Finland. [62]Biocenter Oulu, University of Oulu, Oulu, Finland. [63]Unit of Primary Care, Oulu University Hospital, Oulu, Finland. [64]Center for Occupational and Environmental Medicine, Stockholm County Council, Stockholm, Sweden. [65]College of Veterinary Medicine, Michigan State University, East Lansing, MI, USA. [66]Obstetrics and Gynaecology, University of Helsinki and Helsinki University Hospital, Helsinki, Finland. [67]Medical Research Council Unit The Gambia at the London School of Hygiene and Tropical Medicine, London, UK. [68]Department of Women and Children's Health, King's College London, London, UK. [69]Novo Nordisk Foundation Center for Basic Metabolic Research, Section of Metabolic Genetics, Faculty of Health and Medical Sciences, University of Copenhagen, Copenhagen, Denmark. [70]Department of Public Health, Section of Epidemiology, Faculty of Health and Medical Sciences, University of Copenhagen, Copenhagen, Denmark. [71]Division of Biomedical Informatics and Personalized Medicine, Department of Medicine, University of Colorado Anschutz Medical Campus, Aurora, CO, USA. [72]Department of Public Health & Primary Care, Leuven University, Leuven, Belgium. [73]School of Food Sciences and Nutrition, University of Leeds, Leeds, UK. [74]Department of Medical Sciences, University of Turin, Turin, Italy. [75]AOU Citta della Salute e della Sceinza, CPO Piemonte, Turin, Italy. [76]IMIM (Hospital del Mar Medical Research Institute), Barcelona, Spain. [77]Center for Environmental Research and Children's Health, School of Public Health, University of California, Berkeley, CA, USA. [78]Department of Biological Psychology, Amsterdam Public Health Research Institute, Vrije Universiteit Amsterdam, Amsterdam, The Netherlands. [79]Department of Community and Family Medicine, Duke University Medical Center, Raleigh, NC, USA. [80]The George Institute for Global Health, Nuffield Department of Women's & Reproductive Health, University of Oxford, Oxford, UK. [81]Department of Biochemistry, Université de Sherbrooke, Sherbrooke, QC, Canada. [82]ECOGENE-21 Biocluster, Chicoutimi Hospital, Saguenay, QC, Canada. [83]Laboratory of Precision Environmental Biosciences, Columbia University Mailman School of Public Health, New York, NY, USA. [84]Department of Environmental Health Sciences, Mailman School of Public Health, Columbia University, New York, NY, USA. [85]Division of Research, Kaiser Permanente Northern California, Oakland, CA, USA. [86]Department for Genomics of Common Diseases, School of Public Health, Imperial College London, London, UK. [87]Folkhälsan Institute of Genetics, Helsinki, and Research Programs Unit, Molecular Neurology, University of Helsinki, Helsinki, Finland. [88]Department of Biosciences and Nutrition, Karolinska Institutet, Huddinge, Sweden. [89]School of Basic and Medical Biosciences, King's College London, Guy's Hospital, London, UK. [90]David Hide Asthma and Allergy Research Centre, Isle of Wight, UK. [91]HUSLAB and the Department of Clinical Chemistry, University of Helsinki, Helsinki, Finland. [92]LICAMM, School of Medicine, Univeristy of Leeds, Leeds, UK. [93]Amsterdam Public Health Institute, Vrije Universiteit Amsterdam, Amsterdam, The Netherlands. [94]Center for Epigenetics, Johns Hopkins University School of Medicine, Baltimore, MA, USA. [95]AJ Drexel Autism Institute, Drexel University, Philadelphia, PA, USA. [96]KU Leuven – University of Leuven, Department of Neurosciences, Experimental Neurology and Leuven Institute for Neuroscience and Disease (LIND), Leuven, Belgium. [97]VIB, Center for Brain & Disease Research, Laboratory of Neurobiology, Leuven, Belgium. [98]Department of Mental Health, Bloomberg School of Public Health, Johns Hopkins University, Baltimore, MD, USA. [99]Sachs' Children's Hospital, Stockholm, Sweden. [100]National Institute for Health and Welfare, Helsinki and Oulu, Oulu, Finland. [101]Hospital for Children and Adolescents, Helsinki University Hospital and University of Helsinki, Helsinki, Finland. [102]PEDEGO Research Unit, MRC Oulu, Oulu University Hospital and University of Oulu, Oulu, Finland. [103]Pediatric Allergy and Pulmonology Unit at Astrid Lindgren Children's Hospital, Karolinska University Hospital, Stockholm, Sweden. [104]Department of Population Medicine, Harvard Medical School, Harvard Pilgrim Health Care Institute, Boston, MA, USA. [105]Department of Epidemiology, Colorado School of Public Health, and Department of Pediatrics, University of Colorado School of Medicine, University of Colorado Anschutz Medical Campus, Aurora, CO, USA. [106]Centre for Genetic Origins of Health and Disease, School of Biomedical Sciences, University of Western Australia, Perth, Australia. [107]School of Pharmacy and Biomedical Sciences, Curtin University, Perth, Australia. [108]Department of Global Health and Social Medicine, Harvard Medical School, Boston, MA, USA. [109]Diabetes Unit, Massachusetts General Hospital, Boston, MA, USA. [110]Norwegian Institute of Public Health, Oslo, Norway. [111]Department of Pediatric Oncology and Hematology, Oslo University Hospital, Oslo, Norway. [112]Department of Obstetrics and Gynecology, Duke University Medical Center, Durham, NC, USA. [113]Department of Epidemiology and Biostatistics, University of California, San Francisco, CA, USA. [114]Department of Psychiatry and Behavioral Sciences, Emory University School of Medicine, Altanta, GA, USA. [115]Department of Global Public Health and Primary Care, University of Bergen, Bergen, Norway. [116]Department for Non-Communicable Diseases, Norwegian Institute for Public Health, Oslo, Norway. These authors jointly supervised this work: Debbie A. Lawlor, Caroline L. Relton, Harold Snieder, Janine F. Felix.

