## [Peer Review File · Nature Communications]

Reviewers' comments:

Reviewer #1 (Remarks to the Author):

Kupers *et al.* have performed a meta-analysis EWAS of blood-derived DNA for birth weight. They have identified 955 CpGs associated with this phenotype. Whilst caveats are given in the Discussion and elsewhere in the manuscript, a major criticism is that this paper from the start infers too strong a message regarding the causality of these findings. Particularly, for readers from outside the direct field, who may not be aware of the issues involved. Obvious factors, such as well-known under-reporting of maternal smoking, clearly impact on both birth weight and blood DNA methylation. Their causative implication of the epigenetic changes, as presented, lacks any robust support in this analysis and, therefore, needs revising. My major points are detailed below:

- 1) The inference of these blood DNA methylation changes being potentially causal from the Abstract onwards needs to be excluded or dramatically toned down. No strong data that support the mechanistic causality of these changes is presented. Birth weight is clearly known to be associated with both maternal smoking and BMI. Secondly, robust DNA methylation changes in blood are seen due to smoking and the metabolic abnormalities associated with obesity, such as hyperglycaemia and hypertriglyceridemia. The evidence required for causality should be discussed more precisely throughout the manuscript – and follows similar comments recently made by one of the senior authors of this paper on this issue of causality regarding the recent publication of Toby *et al.*¹
<http://advances.sciencemag.org/content/4/1/eaao4364/tab-e-letters>
- 2) Smoking is found to be a considerable influence, even though it is attempted to be accounted for in the analysis. Therefore, the known external effects on birth weight, such as smoking with decades of evidence² need to be given clearer likelihood of causation in both the phenotype and DNA methylation differences. All the other DNA methylation blood changes need to be interpreted in this light. With smoking this will include subtle inflammatory cell type and proportion changes – potentially not captured by the broad leukocyte category corrections³ - with the further caveat that the cell type correction is not cord-blood specific. Strong overlap with known smoking effects was identified, with 56/955 using the 568 Bonferroni significant results from Joubert *et al.* If the 6,073 smoking-associated CpGs with FDR significance from that study are instead compared, how many further potential changes could be smoking-related?
- 3) This meta-analysis has included many comparatively small genetically diverse cohorts (n < 200). Due to the known strong genetic confounding that exists in DNA methylation array analysis⁴, observed even within Caucasian populations, adding small genetic heterogeneous ancestry groups could reduce power. It would be useful to know whether this broad consortium approach is worthwhile, or whether a more genetically homogenous population selected for more extreme phenotypic differences would in fact be a more powerful approach. The authors performed a European-only analysis and state results were consistent. However, could they comment on this more precisely and if possible quantify this. Additionally, was any available genetic information used to confirm broad ethnicity and exclude outliers?
- 4) Whilst cited in the Introduction, the recent work dissecting out the genetic contribution to birth weight and associated future risk of adult metabolic diseases, which includes many of the authors of this paper, needs to be more clearly stated for those unaware of those findings *i.e.* Horikoshi *et al.*⁵, Beaumont *et al.*⁶ Richmond *et*

*al.*⁷ and the difficulty in interpreting causality in Lawlor *et al.*⁸ *etc.* Furthermore, clarify what the estimates are to how much the genetic component may account for these effects?

- 5) The stated Hypothesis that – “our primary hypothesis was that the intrauterine environment induces epigenetic alterations, which influence fetal growth and hence correlate with birthweight” needs qualification in that the changes investigated for are in cord or heel-pick blood. Therefore, this hypothesis itself is not being robustly tested and biological plausibility is not explained.
- 6) The results, as written in the manuscript, regarding the exclusions of CpGs with significant genetic or technical confounding are not clear to the reader (pg 9, line 289 onwards). Are the 41 CpGs referred to with identified cross-reactive or multimapping probes in the reported 955 CpGs? If so, should they not have been removed prior in QC? Also, the additional 168 CpGs with SNPs in or near (how far?) the CpG site? Were these explored for evidence of genetic confounding either directly by mQTL analysis or by methods such as Gap Hunting.⁴ Were these already removed from the ALSPAC cohort? Considering ~10-20% of probes show potential genetic confounding via techniques, such as Gap Hunting, there is a low level of confounding inferred in ALSPAC from the Dip test for multimodality – is it too conservative?
- 7) Further comment is required on the lack of strong persistence of these changes into older age and their hypothesis of causality. As well as the issue of persisting but decreasing environmental exposures with age, i.e. *in utero* smoking and passive smoking in childhood with reducing parental exposure with age.
- 8) The Mendelian Randomisation analysis findings regarding 3 CpGs and cardiovascular outcomes needs to include caveats regarding: i) horizontal pleiotropy and ii) that if the methylation association is not explicitly disease-related tissue-specific, but found across all tissues, there is reduced confidence in causality.
- 9) The Discussion implies causal relationship without evidence to support this and, therefore, the following sentence needs to be more circumspect – “We observed enrichment of birthweight-associated CpGs among sites that have previously been linked to smoking during pregnancy and pre-pregnancy BMI, consistent with the hypothesis that epigenetic pathways may underlie the observational associations of those prenatal exposures with birthweight.” They are more likely in fact to be driven by smoking and metabolic-related changes in blood cell-type composition.
- 10) The Discussion needs further modification as it is well known that smoking is chronically under-reported by Mothers in pregnancy⁹ – so again it is very unsurprising that smoking effects were identified and should be acknowledged from the start.
- 11) The Discussion regarding gene expression needs to be more nuanced as again it is clear that blood is likely only to be a possible surrogate tissue for any supposed growth phenotype. Furthermore, it needs acknowledging that the DNA methylation to gene expression interpretation is only via association and not backed up by any further biological evidence of mechanism, such as biologically-relevant Transcription Factor binding changes etc. The phrase ‘Proof of principle’ is too strong that these epigenetic changes are causal in birthweight-related CpGs.
- 12) Whilst in the Discussion it is stated “we acknowledge that our main results cannot ascertain causality” – then the Abstract, Introduction and elsewhere should be consistent with this.

13) The final sentence in the Discussion needs modifying from “we cannot exclude the possibility that changes in neonatal blood DNA methylation are caused by variation in fetal growth” to “we cannot exclude the possibility that changes in neonatal blood DNA methylation are caused *by factors that themselves influence fetal growth, such as maternal smoking.*”

1. Tobi, E.W. *et al.* DNA methylation as a mediator of the association between prenatal adversity and risk factors for metabolic disease in adulthood. *Sci Adv* **4**, eaao4364 (2018).
2. Meredith, H.V. Relation between tobacco smoking of pregnant women and body size of their progeny: a compilation and synthesis of published studies. *Hum Biol* **47**, 451-72 (1975).
3. Su, D. *et al.* Distinct Epigenetic Effects of Tobacco Smoking in Whole Blood and among Leukocyte Subtypes. *PLOS ONE* **11**, e0166486 (2016).
4. Andrews, S.V., Ladd-Acosta, C., Feinberg, A.P., Hansen, K.D. & Fallin, M.D. "Gap hunting" to characterize clustered probe signals in Illumina methylation array data. *Epigenetics Chromatin* **9**, 56 (2016).
5. Horikoshi, M. *et al.* Genome-wide associations for birth weight and correlations with adult disease. *Nature* **538**, 248-252 (2016).
6. Beaumont, R.N. *et al.* Genome-wide association study of offspring birth weight in 86 577 women identifies five novel loci and highlights maternal genetic effects that are independent of fetal genetics. *Hum Mol Genet* **27**, 742-756 (2018).
7. Richmond, R.C. *et al.* Using Genetic Variation to Explore the Causal Effect of Maternal Pregnancy Adiposity on Future Offspring Adiposity: A Mendelian Randomisation Study. *PLoS Med* **14**, e1002221 (2017).
8. Lawlor, D. *et al.* *Using Mendelian randomization to determine causal effects of maternal pregnancy (intrauterine) exposures on offspring outcomes: Sources of bias and methods for assessing them [version 1; referees: awaiting peer review]*, (2017).
9. Shipton, D. *et al.* Reliability of self reported smoking status by pregnant women for estimating smoking prevalence: a retrospective, cross sectional study. *BMJ* **339**, b4347 (2009).

Reviewer #2 (Remarks to the Author):

This is an interesting and well-performed study from a large collection of researchers with access to EWAS data from birth cohorts, and experience in the use of the complex analytical methods required for their analysis.

The focus of the paper is on the relationship between DNA sequence methylation (as assessed with the 450K array), early growth (as measured by birthweight), intrauterine exposures (smoking and maternal BMI) and later disease outcomes. The analyses bring together about 9K samples with existing 450K EWAS. The main messages are (a) there are ~955 methylation associations with birthweight; (b) some of these are driven by maternal smoking and BMI; (c) few of them persist into adulthood; (d) these methylation differences are not likely to be driving early growth; and (e) there's only patchy evidence connecting the methylation differences to later traits.

My sense is that these data offer a fairly strong rebuttal of the DOHAD notions that the epidemiological associations between reduced fetal growth and adult disease are mediated through altered methylation. It certainly represents one of the first attempts to bring epigenome-wide analyses to bear on a hypothesis that has historically been dependent on data from animal models (of dubious relevance) and "candidate gene" studies in limited numbers of humans. As the authors point out, sample sizes available inevitably limit the power of the analyses, and prevent definitive statements. As does the fact that they used the 450K array which means they are sampling only a very small, and highly selected subset of the methylation "space" (which is something the authors don't mention).

Nonetheless, this is definitely a step forward in efforts to explore the role of methylation in the nexus of effects connecting early growth to later disease, and the authors are to be commended on the wide-ranging analyses.

The authors have done a good job of describing those complex analyses, but I have some questions and recommendations.

* the discussion is suitably honest about the impact of sample size on the power of the analyses performed and the inferences possible. However, no explicit power calculations are provided, and this should be remedied.

* the authors should describe the limitations of the 450K array.

* on p8 the authors should make clear the tissues that have been analysed (blood it appears from the methods) and the approaches taken to adjust for cellular composition. (The details are in methods, but I think it's preferable for such crucial high level information to be included in results).

* (p9): the authors should describe more clearly what the "SNP effects" they refer to are

* p9: one of the most interesting observations is that the methylation changes seen at birth are not persistent. However, this seems based on an analysis that has different power at different ages, and the inference seems to be based on the number of significant tests. A quick glance at ST6 suggests that coefficients do not seem to decline so much (though confidence intervals get larger). These data seem to merit more sophisticated analyses that take account of the different power of the samples used for each age period.

* (p10): can the authors describe what they mean by "metastable epialleles".

* (p11): I would have expected the authors to offer a little more interpretation of the pathway/GO analyses in ST11. In the discussion they are rather dismissive of these results, yet here the text

seems more optimistic (but still superficial).

* (p11): The MR analyses seem to have been performed using single SNPs, and there has been no attempt to explore the assumptions of the MR method (eg wrt pleiotropy). I wonder also why the authors chose not to employ a GRS approach to extend power (and also overcome some pleiotropy concerns).

* (p13): the section in the discussion on the cis-eQTM results seems to assume a causal direction from methylation to expression, but this seems naive based on other data (eg Kilpinen et al). As these analyses do not involve genetic variants, there is no easy way to infer the direction of causality here

* (p13): I think its a mistake to describe any study of 450K data as "comprehensive" given the limited extent of coverage.

* Fig 4: great figure that summarises a lot of data. It wasnt immediately obvious what the dots represented (I assume all the dots in the orange track are the 955 BW associated CpGs?).

Reviewer #3 (Remarks to the Author):

Review of Birthweight paper

Summary: This is a clear, well written manuscript with extensive analytical depth. The authors evaluate the association between birthweight and DNA methylation in the PACE consortium, including 24 independent cohorts in this meta-analysis. The expertise and experience in this group is well suited to tackle this important research question. They hypothesize that birthweight may impact later life health outcomes through epigenetic mechanisms and compare their results with other findings from PACE, including smoking during pregnancy and pre-pregnancy BMI studies. This paper extends from earlier PACE work by including more cohorts, incorporating mendelian randomization, additional public databases, and more thorough analyses among study participants across life stage. All comments below are minor and for the authors' consideration to potentially improve clarity for the reader.

The statistical analyses are all appropriate, and replicate methods used in previous work from the PACE consortium, while adding depth to functional and causal approaches. This work is of great interest to the scientific communities interested in early life exposures and long-term health implications, as well as the underlying mechanisms explaining these associations.

Title: "Meta-analysis of epigenome-wide association studies in neonates reveals widespread differential DNA methylation associated with birthweight." The title does not reflect the longitudinal aspects of this work which I find to be rather innovative, despite limited conclusive findings. Authors could consider minor modification to the title to reflect the approach.

Causal analyses: The authors note the following, starting on line 355:

"This large-scale meta-analysis shows that birthweight is associated with widespread differences in DNA methylation. We observed enrichment of birthweight-associated CpGs among sites that have previously been linked to smoking during pregnancy²¹ and pre-pregnancy BMI²², consistent with the hypothesis that epigenetic pathways may underlie the observational associations of those prenatal exposures with birthweight^{17,19,45}."

-Separately, the authors used MR to explore causal association of methylation at identified CpGs and birthweight as well as later-life health outcomes. How was maternal smoking during pregnancy incorporated into these models? Could they address their statement above in a more direct analysis such as MR by evaluating the causal pathway across maternal smoking \diamond CpGs \diamond

birth weight? Or maternal smoking + pre-pregnancy BMI \diamond CpG methylation \diamond birth weight?

Effect measure modification by race/ethnicity: The authors note that the results were consistent when restricting to just European ancestry study participants (Supp Fig 2, blue track in Fig 4). What do they find when restricted to African ancestry study participants only and/or Hispanic ancestry only? Even though the sample size/statistical power is dramatically reduced (just NEST and Healthy Start for AA cohorts), it would be helpful to indicate whether there is consistent direction of effect for top CpGs. This could be a sensitivity analysis in the supplement. Alternatively, they could test for effect measure modification of the association between methylation and birthweight by race/ethnicity for top 1-5 or so CpGs. And given the imbalance in sample size in European vs. non European ancestry, they could consider a random sample of the # of AA subjects among the Europeans so the analyses are not swayed by sample size alone. E.g 1000 Europeans and 1000 AAs in one of the proposed sensitivity analyses. Overall, these suggestions are a broad request for the authors to add some analytical component to further address whether the findings are consistent across race/ethnicity, rather than potentially sweep that aspect under a sample size rug, if you will. This would be beneficial for the research community looking to replicate the presented approaches.

Consistency of effects across life stage: Starting on line 296: "In childhood (2-13y; 2,756 children from 10 studies), adolescence (16-18y; 2,906 adolescents from 6 studies) and adulthood (30-45y; 1,616 adults from 3 studies), we observed 91, 51 and 44 of the 955 CpGs, respectively, to be nominally associated with birthweight ($p < 0.05$), with consistent directions of association. Eleven CpGs showed differential methylation across all 4 age periods."
-Was the direction of effect for these CpGs consistent?

MR, starting on line 338: " To explore their causal associations with birthweight and 139 complex later-life outcomes (Supplementary Table 12), we used 135 local methylation quantitative trait loci (cis-mQTLs), genetic variants associated with methylation levels, using a publicly available mQTL database⁴³ as instrumental variables for 127 of the 955 birthweight-associated CpGs in two-sample MR. For the remaining CpGs (i.e. 828 [87%]) no genetic instrumental variables could be identified in the publicly available mQTL database. Hence, we could not conduct MR for those CpGs."
--Don't these cohorts also have GWAS data? Why were genetic variants not used in the MR approaches?

Confounding by cell type, lines 397-401: The authors addressed the issue of potential confounding by cell type by adjusting for cell type composition in their analyses. They note that they used the adult reference panel which had now been improved by the availability of cord blood reference panels. They conduct a sub analysis using one of the updated cord panels but did not rerun the entire EWAS with this updated reference. This seems appropriate and adequately addresses the issue, given the extensive analysis involved in these types of meta-analyses.
Relevant excerpt: "DNA methylation varies 397 between leukocyte subtypes⁴⁹ and we used an adult whole blood reference to correct for this in our main analyses^{50,51}, as our study-specific analyses were completed before the widespread availability of specific cord blood reference datasets^{52,53}. However, we observed very similar findings in two studies (Generation R and GECKO) when we compared the results with those using one of the currently available cord blood references⁵²."

Consideration of DAGs: Figure 1 displays hypothetical paths. Given the number of epidemiologists involved in this consortium, modification of the figure to display a proper DAG could be given some consideration, including potential confounders. This could be provided in a supplemental figure rather than replacing Figure 1, which provides the best conceptual overview for the paper. I am not suggesting a modification to their analyses based on what the DAG looks like, but to consider clearer integration of some epidemiologic methods in this type of omics work, either here or in the future.

Gambia study: The authors acknowledge the study from the Gambia but this is not included as one of the cohorts in the meta-analysis (e.g., not listed in Table 1). It appears only relevant to the MR analyses which appears to be online publicly accessible data. Inclusion of EWAS data from this cohort, if available, would have added some valuable diversity to the study population. In the acknowledgements, it looks like the cohort data was involved.

Reviewers' comments:

Reviewer #1 (Remarks to the Author):

Kupers et al. have performed a meta-analysis EWAS of blood-derived DNA for birth weight. They have identified 955 CpGs associated with this phenotype. Whilst caveats are given in the Discussion and elsewhere in the manuscript, a major criticism is that this paper from the start infers too strong a message regarding the causality of these findings. Particularly, for readers from outside the direct field, who may not be aware of the issues involved. Obvious factors, such as well-known under-reporting of maternal smoking, clearly impact on both birth weight and blood DNA methylation. Their causative implication of the epigenetic changes, as presented, lacks any robust support in this analysis and, therefore, needs revising. My major points are detailed below:

- 1. The inference of these blood DNA methylation changes being potentially causal from the Abstract onwards needs to be excluded or dramatically toned down. No strong data that support the mechanistic causality of these changes is presented. Birth weight is clearly known to be associated with both maternal smoking and BMI. Secondly, robust DNA methylation changes in blood are seen due to smoking and the metabolic abnormalities associated with obesity, such as hyperglycaemia and hypertriglyceridemia. The evidence required for causality should be discussed more precisely throughout the manuscript – and follows similar comments recently made by one of the senior authors of this paper on this issue of causality regarding the recent publication of Toby et al.¹**

We agree with the reviewer that the associations that were found between birth weight and DNA methylation should be interpreted with caution. Throughout the revised manuscript, including abstract, background, results and discussion we have now been more cautious in our interpretation of these results and more explicit in stating that they are associational and not necessarily causal.

We aimed *a priori* to perform Mendelian randomization (MR) analyses to further explore causation. However, as explained in the original submission, we were only able to identify mQTLs as instrumental variables for a minority (n=127 (13%)) of the 955 CpG sites that were differentially methylated. In addition, for 109 (86%) of those 127 differentially methylated CpGs, only one mQTL was available that could be used as a genetic instrumental variable. This meant that we were unable to perform MR or distinguish any causal effects from bias due to horizontal pleiotropy for the majority (n=937, 98%) of the CpGs, as this is not possible with ≤ 1 mQTL and has limited reliability with only 2 or 3 mQTLs. In the revised manuscript we now acknowledge that

we set out to undertake the MR analyses, provide our results of the mQTLs found for 127 of the differentially methylated CpGs (and note that for 87% we could not find any mQTLs to use as instrumental variables) and then acknowledge that the small proportion of differentially methylated CpGs with potential genetic instruments (mQTLs) and the fact that of those most had only one mQTL strongly limit our ability to produce results that would be interpretable in terms of causality. In line with this, we no longer provide MR results, as we feel that there was too little support to draw firm conclusions on causality.

We have changed the following sections of the manuscript and added **Supplementary Table 15** (please find below the text in red for the relevant changes, because of the many textual changes we decided to only upload a clean manuscript file without track changes):

Background

To aid functional interpretation we: (i) explored the overlap of identified cytosine-phosphate-guanine sites (CpGs) that were differentially methylated in relation to birthweight with those known to be associated with intrauterine exposure to smoking, famine and different levels of BMI and folate; (ii) associated DNA methylation at identified CpGs with gene expression and (iii) aimed to explore potential causal links with birthweight and later-life health using Mendelian randomization (MR)²². (Pages 7-8, lines 40-44)

Results

*We aimed to explore causality using MR analysis, in which genetic variants associated with methylation levels (methylation quantitative trait loci (mQTLs)) are used as instrumental variables to appraise causality. For 828 (87%) of the 955 birthweight-associated CpGs, no mQTLs were identified in a publicly available mQTL database³⁹. For 109 (85.8%) of the remaining 127 CpGs, only 1 mQTL was identified and for the remainder none had more than four mQTLs (**Supplementary Table 15** provides a complete list of all mQTLs identified for these 127 CpGs). Many of the currently available methods that can be used as sensitivity analyses to explore whether MR results are biased by horizontal pleiotropy (a single mQTL influencing multiple traits) require more than one genetic instrument (here mQTLs) and even with two or three this can be difficult to interpret⁴³. Having determined that it was not possible to undertake MR analyses of 87% of the birthweight-related differentially methylated CpGs (because we did not identify any mQTLs), and for the majority of the remaining CpGs we would not have been reliably able to distinguish causality from horizontal pleiotropy (because only 1 mQTL could be identified), we decided not to pursue MR analyses further. (Pages 13-14, lines 173-185)*

Discussion

We originally aimed to explore this using MR analysis. For the vast majority of the birthweight-associated CpGs, no genetic instrumental variables were available. For the remaining 127 CpGs, only 1 mQTL was available, which would make it impossible to disentangle causality from horizontal pleiotropy. To ensure a strong basis for future MR analyses on this topic, there is a clear need for a more extensive mQTL resource. (Page 16, lines 232-236)

- 2. Smoking is found to be a considerable influence, even though it is attempted to be accounted for in the analysis. Therefore, the known external effects on birth weight, such as smoking with decades of evidence² need to be given clearer likelihood of causation in both the phenotype and DNA methylation differences. All the other DNA methylation blood changes need to be interpreted in this light. With smoking this will include subtle inflammatory**

cell type and proportion changes – potentially not captured by the broad leukocyte category corrections³ - with the further caveat that the cell type correction is not cord-blood specific.

Strong overlap with known smoking effects was identified, with 56/955 using the 568 Bonferroni significant results from Joubert et al. If the 6,073 smoking-associated CpGs with FDR significance from that study are instead compared, how many further potential changes could be smoking-related?

We agree that smoking could explain (i.e. confound) associations between DNA methylation and birthweight. In the main analyses all studies adjusted for maternal smoking, although we acknowledge that residual confounding could remain. We now discuss more explicitly throughout the paper that the associations we observe could partly be explained by residual confounding and that they may not be causal.

In the revised manuscript we now present additional analyses in which we look at the extent of overlap of the 8,696 FDR hits from our EWAS of birthweight with the 6,073 FDR hits for smoking, 16,067 FDR hits for maternal BMI (which were sent to us upon request by the corresponding author of that study), 443 FDR hits for folate, and 7 FDR hits for famine. The proportion of overlap was broadly similar to that found when comparing Bonferroni corrected p-values. We made the following changes to the Results:

Results

We did not find evidence for overlap with plasma folate³¹, and no famine exposure related CpGs were previously presented at the Bonferroni-corrected level of $P < 1.06 \times 10^{-7}$ ³². In additional analyses for overlap between all FDR hits from the birthweight EWAS with those FDR hits presented in the smoking, maternal BMI, folate and famine EWAS, we found an overlap of 439/8,696 CpGs (5.0%, $p_{\text{enrichment}} < 5 \times 10^{-324}$) for smoking, 625/8,696 CpGs (7.2%, $p_{\text{enrichment}} = 3.74 \times 10^{-67}$) for maternal BMI and 15/8,696 (0.2%, $p_{\text{enrichment}} = 0.02$) for folate. For famine we did not observe overlap. (page 12, lines 127-132)

Methods

No CpGs reached the Bonferroni-corrected cut-off for famine³². We additionally appraised this overlap using the $FDR < 0.05$ cut-off for all traits ($n = 8,696$ birthweight-related CpGs, $n = 6,703$ smoking-related CpGs, $n = 16,067$ BMI-related CpGs, $n = 443$ folate-related CpGs, $n = 7$ famine-related CpGs). These FDR results were available from the publications for smoking, folate and famine, and we obtained them from the corresponding author for BMI. (pages 26-27, lines 432-436)

The reviewer suggests that residual confounding by smoking may also occur because of the potential impact of maternal smoking on subtle inflammatory responses and hence white cell proportions that may not be fully captured by the adjustment for broad white blood cell categories. We now note this possibility in the revised discussion. In the original manuscript we explained the fact that we did not use the cord blood-specific reference in the Discussion. There we also explained that, in two relatively large cohorts included in the meta-analysis, we reran the main analysis using the more accurate cord-blood specific cell type correction and we did not find notable differences in that analysis as compared to the analysis presented in the paper using the adult reference. We made the following changes to the Discussion of the manuscript:

*The overlap that we observed with pregnancy smoking related CpGs may reflect the possibility that smoking-related CpGs capture smoking better than self-report^{46,47}, in line with expectations of pregnant women underreporting their smoking behaviour. Adjustment for maternal smoking and BMI may have masked a greater level of overlap between our results and EWAS of these two maternal exposures. The fact that we find an association of DNA methylation across the genome with birthweight provides some support for our conceptual framework shown in **Figure 1**. However, we acknowledge that the associations that we have observed may also be explained by causal effects of maternal pregnancy exposures on both DNA methylation and fetal growth, as well as subtle inflammatory responses in cell type proportions associated with maternal smoking that might not have been completely captured with the currently available cell type estimation methods. (page 15, lines 192-202)*

- 3. This meta-analysis has included many comparatively small genetically diverse cohorts (n < 200). Due to the known strong genetic confounding that exists in DNA methylation array analysis⁴, observed even within Caucasian populations, adding small genetic heterogeneous ancestry groups could reduce power. It would be useful to know whether this broad consortium approach is worthwhile, or whether a more genetically homogenous population selected for more extreme phenotypic differences would in fact be a more powerful approach. The authors performed a European-only analysis and state results were consistent. However, could they comment on this more precisely and if possible quantify this. Additionally, was any available genetic information used to confirm broad ethnicity and exclude outliers?**

We agree with the reviewer that it is important to test heterogeneity in study samples that were included in the meta-analysis. For this reason we chose to exclude from the downstream analyses any associations with evidence of between-study heterogeneity, defined *a priori* as $I^2 > 50$. In the revised manuscript we have also added 'leave one out' analyses in which we have rerun the main EWAS meta-analysis repeatedly with one of the 24 studies removed each time and explore whether any single study is inconsistent with others. We are unable to determine ethnicity using genetic information or adjust for it using this information in the PACE consortium as not all contributing studies have genetic data. In the revised manuscript we have further clarified how we defined ethnicity and how we compared the main meta-analyses results (all ethnicities combined) with those from Europeans, African Americans and Hispanics (see pages 24-25). We have made the following changes to the Results section:

*Findings were consistent with results from our main analyses when restricted to participants of European ethnicity, with a Pearson correlation coefficient for effect estimates of 0.99 for the 955 birthweight-associated CpGs (**Supplementary Figure 2**, blue track (2) in **Figure 4** and **Supplementary Table 3**) and 0.90 for all 450k CpGs. Comparing the main meta-analyses to the four Hispanic cohorts and the two African cohorts revealed that 94.8% and 74.2% of the 955 CpGs showed consistent direction of association, with Pearson correlation coefficients for point estimates of 0.82 and 0.48, respectively (**Supplementary Table 3**). In leave-one-out analyses, in which we reran the main meta-analysis repeatedly with one of the 24 studies removed each time, there was no strong evidence that any one study influenced findings consistently across the 955 differentially methylated CpGs that passed Bonferroni correction and for which between study heterogeneity had an $I^2 < 50\%$. For 151/955 CpGs (15.8%) the difference in mean birth weight for a 10% greater methylation at that site varied by $\geq 20\%$ or more with removal of a study, but the study resulting in the change was different for different CpGs.*

Supplementary Figures 3.1-3.20 show the results for a random 10 plots where removal of one study changed the result by 20% or more and a random 10 where this was not the case; full results are available on request from the authors. (page 10, lines 71-85)

We now also briefly discuss the comparison of ethnicities in the Discussion:

The majority of participants were of European ethnicity and when analyses were restricted to those of European ethnicity the results were essentially identical to those with all studies included. Direct comparisons of the main analysis with analyses in those of Hispanic or of African ethnicity for the 955 hits suggested strong correlations with Hispanic but weaker with African ethnicity. However, these results need to be treated with caution. First, we had very few studies of Hispanic and African populations. Second, we only compared the initial hits from the main meta-analysis with all ethnicities included. A detailed exploration of ethnic differences would require similar large samples for each ethnic group and within ethnic EWAS, which is beyond the scope of the data currently available. (page 18, lines 266-274)

And we mention the definition of ethnicity in the Methods:

*Ethnicity was defined using maternal or self-report, unless specified otherwise in study-specific **Supplementary Methods**. (page 25, lines 411-413)*

4. Whilst cited in the Introduction, the recent work dissecting out the genetic contribution to birth weight and associated future risk of adult metabolic diseases, which includes many of the authors of this paper, needs to be more clearly stated for those unaware of those findings i.e. Horikoshi et al.⁵, Beaumont et al.⁶ Richmond et al.⁷ and the difficulty in interpreting causality in Lawlor et al.⁸ etc.

Furthermore, clarify what the estimates are to how much the genetic component may account for these effects?

We have extended the Methods and Results sections of the manuscript so that we now compare results from our analyses with the most recent GWAS of fetal genotype associations with birthweight (Horikoshi et al.) and that of maternal genotype associations with birthweight (Beaumont et al.):

Results

*To compare these EWAS results to those from genetic studies, we used the 60 recently published fetal SNPs associated with birthweight in a GWAS meta-analysis of 153,781 newborns³⁷ and mapped the CpG sites identified in the EWAS to these SNPs to seek evidence of co-localisation of genetic and epigenetic variation (**Supplementary Table 10**). We repeated this for the 10 recently published maternal SNPs associated with birthweight in a GWAS meta-analysis of 86,577 women³⁸ (**Supplementary Table 11**). We observed that one or more of the 955 birthweight-associated CpGs were within 4Mb (+/- 2Mb) of 35/60 fetal and all 10 maternal birthweight-associated SNPs were within 4Mb (+/- 2Mb) of one or more of the 955 birthweight-associated CpGs. Of the 35 fetal SNPs, three were located in the same gene as the CpG, as was one of the ten maternal SNPs. Ten fetal and four maternal SNPs were within 100kb of identified CpGs. In a look-up of the fetal and maternal SNPs from GWAS of birthweight in an online cord blood methylation quantitative trait loci (mQTL) database (mqtl.db.org³⁹), 35 fetal and 4 maternal SNPs affected methylation at some CpG(s), but none at the 955 birthweight-associated CpGs specifically. (pages 12-13, lines 144-156)*

Methods

We compared the birthweight-associated CpGs with the 60 SNPs from the most recent GWAS meta-analyses of fetal genotype associations with birthweight in >150,000 newborns³⁷ and with 10 SNPs from

the most recent GWAS meta-analysis of maternal genotype associations with birthweight in >86,000 women³⁸. With this comparison we checked if the EWAS top hits were located within a 4Mb window (+/- 2Mb) surrounding these SNPs. We additionally checked whether SNPs and CpGs were located in the same gene. (page 27, lines 449-454)

- 5. The stated Hypothesis that – “our primary hypothesis was that the intrauterine environment induces epigenetic alterations, which influence fetal growth and hence correlate with birthweight” needs qualification in that the changes investigated for are in cord or heel-pick blood. Therefore, this hypothesis itself is not being robustly tested and biological plausibility is not explained.**

We agree with the reviewer that biological plausibility of this hypothesis is not directly tested in our study. The primary hypothesis as presented in the manuscript was intended to represent the overall conceptual framework more so than the specific hypothesis tested. We have rephrased this part of the Background:

Our overall conceptual framework in this study was that the intrauterine environment induces epigenetic alterations, which influence fetal growth and hence correlate with birthweight. For this study, we hypothesised that there are associations between DNA methylation and birthweight. We further aimed to explore if these epigenetic alterations are associated with later disease outcomes (Figure 1). (page 7, lines 23-27)

- 6. The results, as written in the manuscript, regarding the exclusions of CpGs with significant genetic or technical confounding are not clear to the reader (pg 9, line 289 onwards). Are the 41 CpGs referred to with identified cross-reactive or multimapping probes in the reported 955 CpGs? If so, should they not have been removed prior in QC? Also, the additional 168 CpGs with SNPs in or near (how far?) the CpG site? Were these explored for evidence of genetic confounding either directly by mQTL analysis or by methods such as Gap Hunting.⁴ Were these already removed from the ALSPAC cohort? Considering ~10-20% of probes show potential genetic confounding via techniques, such as Gap Hunting, there is a low level of confounding inferred in ALSPAC from the Dip test for multimodality – is it too conservative?**

The 41 and 168 CpGs are among the 955 birthweight-related CpGs. We chose to flag rather than exclude these CpGs, because they may represent potentially interesting biological signals, but we did want to make the readers aware of potential issues in the interpretation of the results for these probes. Whether probes with potential SNP effects are associated with SNPs in our populations depends on the frequency of the SNPs in the included populations, which is not known for all included studies. We chose to include all probes in our main meta-analysis, which is in accordance with the analysis strategy suggested in the Gap Hunter paper (Andrews *et al.* 2016). That paper concludes that as there are multiple possible drivers of bi-modal DNA methylation signals, most probes should be retained but potential gap probes should be 'flagged' and explored in greater

detail *post hoc*. We used the dip test, rather than Gap Hunting, but both explore the presence of dips/gaps (i.e. multimodality) in the methylation distribution of each of the potentially problematic CpGs and they have a comparable theoretical underpinning. We specifically tested this in the 168 potential polymorphic CpGs in ALSPAC and with the dip test we found no evidence for genetic confounding. We now make this clearer in the Results section:

*We identified that 41 of the 955 differentially methylated CpGs co-hybridised to alternate sequences (i.e. cross-reactive sites; **Supplementary Table 6**). For these we cannot distinguish whether the differential methylation is at the locus that we have reported or one that the probe cross-reacts with. We also identified that 168 of the 955 differentially methylated CpGs potentially contained a single-nucleotide polymorphism (SNP) at cytosine or guanine positions (i.e. polymorphic CpGs; **Supplementary Table 6**). Polymorphic CpGs may affect probe binding and hence measured DNA methylation levels^{26,27}. We used one of the largest studies (ALSPAC; N=633) to explore this. We found no indication of bimodal distributions for any of the 168 CpGs suggesting SNPs had not markedly affected methylation measurements at these sites (dip test p-values: 0.299 to 1.00)²⁸⁻³⁰. (pages 10-11, lines 93-101)*

We further added a sentence to the Discussion:

We emphasise interpretation with caution for the CpGs that we flagged as potentially cross-reactive, as the measured methylation levels may represent methylation at either of the potential loci. Also, although we did not find evidence for polymorphic effects for the 168 potentially polymorphic CpGs in ALSPAC, we cannot completely exclude these potential polymorphic effects in the meta-analysed results. (pages 17-18, lines 262-266)

- 7. Further comment is required on the lack of strong persistence of these changes into older age and their hypothesis of causality. As well as the issue of persisting but decreasing environmental exposures with age, i.e. in utero smoking and passive smoking in childhood with reducing parental exposure with age.**

We now discuss the lack of strong persistence in more detail and have added the potential issue of decreasing environmental exposures with age to the Discussion:

The differential methylation associated with birthweight in neonates persisted only minimally across childhood and into adulthood. Larger (preferably longitudinal) studies are needed to explore persistent differential methylation in more detail and with better power at older ages. It is possible that inclusion of the Gambia study in the childhood EWAS (which was the only non-European study in these analyses and was not included in the main meta-analyses with neonatal blood) might have impacted these results, although this study made up just 7% of the total child follow-up sample. A rapid attenuation of differential methylation in relation to birthweight in the first years after birth has previously been reported²⁰, but our sample size for these analyses may have been too small to detect persistence. This rapid decrease, if real, may indicate a reduction in the 'dose' of the child's exposure to maternal factors such as smoking once the offspring is delivered, with that reduction continuing as the child ages. Persistence of birthweight-related differential DNA methylation may not necessarily be a prerequisite for long-term effects, as transient differential methylation in early life may cause lasting functional alterations in organ structure and function that predispose to later adverse health effects. (pages 15-16, lines 203-215)

- 8. The Mendelian Randomisation analysis findings regarding 3 CpGs and cardiovascular outcomes needs to include caveats regarding: i) horizontal pleiotropy and ii) that if the methylation association is not explicitly disease-related tissue-specific, but found across all tissues, there is reduced confidence in causality.**

As described in our reply to the first comment, we have now decided to no longer provide any MR results but explain our initial intention to do so and discuss our reasoning for ultimately not including the results in this manuscript. We nevertheless thank the reviewer for highlighting these relevant issues.

- 9. The Discussion implies causal relationship without evidence to support this and, therefore, the following sentence needs to be more circumspect – “We observed enrichment of birthweight-associated CpGs among sites that have previously been linked to smoking during pregnancy and pre-pregnancy BMI, consistent with the hypothesis that epigenetic pathways may underlie the observational associations of those prenatal exposures with birthweight.” They are more likely in fact to be driven by smoking and metabolic-related changes in blood cell-type composition.**

We agree with the reviewer that we can only very carefully interpret these results in terms of causality, and we should keep in mind confounding possibilities. We therefore refer to the reviewer’s point 2 where we present the adjusted text in the manuscript.

- 10. The Discussion needs further modification as it is well known that smoking is chronically under-reported by Mothers in pregnancy⁹ – so again it is very unsurprising that smoking effects were identified and should be acknowledged from the start.**

We agree that smoking is likely to be underreported by mothers during pregnancy, and have included comments pertaining to this in the beginning of the Discussion, as suggested by the reviewer (see page 14 and our response to the reviewer’s comment 2).

- 11. The Discussion regarding gene expression needs to be more nuanced as again it is clear that blood is likely only to be a possible surrogate tissue for any supposed growth phenotype. Furthermore, it needs acknowledging that the DNA methylation to gene expression interpretation is only via association and not backed up by any further biological evidence of mechanism, such as biologically-relevant Transcription Factor binding changes etc. The phrase ‘Proof of principle’ is too strong that these epigenetic changes are causal in birthweight-related CpGs.**

We have now toned down the wording according to the reviewer’s suggestions:

Methylation is known *to be associated with* gene expression⁴⁸. However, we found no consistent associations between birthweight-related methylation and gene expression in two childhood studies. This could be due to the relatively small sample sizes, differences in ethnicities, age, or platforms to measure gene expression. The use of blood, *which is likely only a possible surrogate tissue for fetal growth phenotypes*, for gene expression analysis might also explain the lack of findings. We did find multiple cis-eQTMs among the birthweight-related CpGs at which methylation was related to gene expression in blood when using a publicly available database from a larger adult sample⁴⁰, *providing some evidence that birthweight-related differentially methylated CpGs may be associated with gene expression. These initial in silico association analyses need further exploration to establish any underlying causal mechanisms.* (page 16, lines 216-225)

12. Whilst in the Discussion it is stated “we acknowledge that our main results cannot ascertain causality” – then the Abstract, Introduction and elsewhere should be consistent with this.

We have now adjusted this throughout the manuscript, according to the above suggestions by the reviewer.

13. The final sentence in the Discussion needs modifying from “we cannot exclude the possibility that changes in neonatal blood DNA methylation are caused by variation in fetal growth” to “we cannot exclude the possibility that changes in neonatal blood DNA methylation are caused by factors that themselves influence fetal growth, such as maternal smoking.”

We have now modified this sentence:

*That is, whilst we have hypothesised that variation in fetal DNA methylation influences fetal growth and hence birthweight, and undertaken the analyses accordingly, we cannot exclude the possibility that differences in neonatal blood DNA methylation are caused by variation in fetal growth itself, or that the association is confounded by factors, including maternal smoking and BMI, that independently influence both fetal growth and DNA methylation (as suggested in **Figure 1**). (page 17, lines 255-260)*

Reviewer #2 (Remarks to the Author):

This is an interesting and well-performed study from a large collection of researchers with access to EWAS data from birth cohorts, and experience in the use of the complex analytical methods required for their analysis.

The focus of the paper is on the relationship between DNA sequence methylation (as assessed with the 450K array), early growth (as measured by birthweight), intrauterine exposures (smoking and maternal BMI) and later disease outcomes. The analyses bring together about 9K samples with existing 450K EWAS. The main messages are (a) there are ~955 methylation associations with birthweight; (b) some of these are driven by maternal smoking and BMI; (c) few of them persist into adulthood; (d) these methylation differences are not likely to be driving early growth; and (e) there's only patchy evidence connecting the methylation differences to later traits.

My sense is that these data offer a fairly strong rebuttal of the DOHAD notions that the epidemiological associations between reduced fetal growth and adult disease are mediated through altered methylation. It certainly represents one of the first attempts to bring epigenome-wide analyses to bear on a hypothesis that has historically been dependent on data from animal models (of dubious relevance) and "candidate gene" studies in limited numbers of humans. As the authors point out, sample sizes available inevitably limit the power of the analyses, and prevent definitive statements. As does the fact that they used the 450K array which means they are sampling only a very small, and highly selected subset of the methylation "space" (which is something the authors don't mention).

Nonetheless, this is definitely a step forward in efforts to explore the role of methylation in the nexus of effects connecting early growth to later disease, and the authors are to be commended on the wide-ranging analyses.

The authors have done a good job of describing those complex analyses, but I have some questions and recommendations.

1. the discussion is suitably honest about the impact of sample size on the power of the analyses performed and the inferences possible. However, no explicit power calculations are provided, and this should be remedied.

We acknowledge the importance of power in studies of this type and, although bound by the total sample size available to us when embarking upon this study, we have now added a *post-hoc* power calculation for our primary analysis to the Discussion section. (please find below the text in red for the relevant changes, because of the many textual changes we decided to only upload a clean manuscript file without track changes)

In a post-hoc power calculation based on the sample size of 8,825 with a weighted mean birthweight of 3560g (weighted mean standard deviation (SD): 483g) and with an alpha set at the Bonferroni-corrected level of $P < 1.06 \times 10^{-7}$ we had 80% power, with a two-sided test, to detect a minimum difference of 0.13 SD (63 grams) in birth weight for each SD increase in methylation. The difference in methylation corresponding to a 1 SD increase differs per CpG, as it depends on the distribution of the methylation values. We acknowledge that smaller differences which might be clinically or biologically relevant may not have been identified in the current analysis. Nonetheless, to our knowledge this analysis has brought together all studies currently available with relevant data and is the largest published study of this association. (page 17, lines 238-246)

2. the authors should describe the limitations of the 450K array.

We now described the limited coverage and the selected subset of CpGs in the Discussion:

The 450k array that was used to measure genome-wide DNA methylation only covers 1.7% of the total number of CpGs present in the genome and specifically targets CpGs in promoter regions and gene bodies⁵². (page 17, lines 260-262)

- 3. on p8 the authors should make clear the tissues that have been analysed (blood it appears from the methods) and the approaches taken to adjust for cellular composition. (The details are in methods, but I think its preferable for such crucial high level information to be included in results).**

We now specify the tissue type in the "Meta-analysis" paragraph. We additionally thank the reviewer for noticing that the approach to adjust for cellular composition, although mentioned in the methods, was missing from the results section. We therefore included this information in the second sentence of the "Meta-analysis" paragraph:

Methylation at 8,696 CpGs, measured in neonatal blood using the Illumina Infinium® HumanMethylation450 BeadChip assay and adjusted for cell type heterogeneity²⁹⁻³¹, was associated with birthweight ... (page 9, lines 55-57)

- 4. (p9): the authors should describe more clearly what the "SNP effects" they refer to are**

We have now clarified this in the Results (please see our response to reviewer 1, point 6)

- 5. p9: one of the most interesting observations is that the methylation changes seen at birth are not persistent. However, this seems based on an analyses that have different power at different ages, and the inference seems to be based on the number of significant tests. A quick glance at ST6 suggests that coefficients do not seem to decline so much (though confidence intervals get larger). These data seem to merit more sophisticated analyses that take account of the different power of the samples used for each age period.**

We now additionally present correlations between the effect estimates for the 955 CpGs at the different ages, which indicate attenuation of associations, even though the direction of associations on the whole do not change. We have changed this in the Results:

Of the 955 CpGs, 57%, 53% and 54% CpGs showed consistency in direction of association in childhood, adolescence and adulthood, but these 955 CpGs were only weakly correlated with methylation levels in neonatal blood (Pearson correlation coefficients 0.16, 0.04 and 0.03, respectively for methylation level correlations with neonatal blood for blood taken in childhood, adolescence and adulthood). (page 11, lines 114-118)

We agree with the reviewer that the smaller sample size at older ages is a limitation of the analyses of persistence of the signals beyond birth and discuss this matter and potential limitations in the Discussion. Please find the text in our response to reviewer 1, point 7.

- 6. (p10): can the authors describe what they mean by "metastable epialleles".**

In the Results we explained metastable epialleles as “*metastable epialleles (loci for which the methylation state is established in the periconceptual period^{33,34}).*” (page 12, lines 135-136). We now added additional explanation and a relevant citation, which can be found in the Methods:

The metastable epialleles were derived from a recently published study that identified 2,210 putative metastable epialleles³⁴. (page 27, lines 440-441)

- 7. (p11): I would have expected the authors to offer a little more interpretation of the pathway/GO analyses in ST11. In the discussion they are rather dismissive of these results, yet here the text seems more optimistic (but still superficial).**

We have now added the following text regarding the results of the GO analyses in the Discussion:

Functional enrichment analyses showed inconclusive results in a range of pathways, with the most significantly enriched pathways including DNA binding, hematopoiesis, and immune, skeletal, and cardiovascular system development. (page 16, lines 227-229)

- 8. (p11): The MR analyses seem to have been performed using single SNPs, and there has been no attempt to explore the assumptions of the MR method (eg wrt pleiotropy). I wonder also why the authors chose not to employ a GRS approach to extend power (and also overcome some pleiotropy concerns).**

The reviewer is correct in saying that the use of single SNPs in our MR analysis is a clear limitation. In response to this comment and similar comments from reviewer 1, we decided to remove MR analyses from the manuscript but explain our initial intention of performing an MR analysis and then discuss our reasoning for ultimately not including this. See our response to reviewer 1, point 1 for full details of our reasoning and the changes we have made to the manuscript in relation to this.

- 9. (p13): the section in the discussion on the cis-eQTM results seems to assume a causal direction from methylation to expression, but this seems naive based on other data (eg Kilpinen et al). As these analyses do not involve genetic variants, there is no easy way to infer the direction of causality here**

Indeed, the wording in the discussion section seemed to imply this, although we are aware that the relation between DNA methylation and gene expression is more complex. We have adjusted the wording as follows:

Methylation is known to be associated with gene expression⁴⁸. (page 16, line 216)

10. (p13): I think it's a mistake to describe any study of 450K data as "comprehensive" given the limited extent of coverage.

We now describe the coverage of the 450k array as a limitation in the Discussion section, as can be read in our response to reviewer 2, point 2. In addition, we have changed "comprehensive" to "extensive" in the Discussion section to emphasize the various analyses included in the manuscript:

*Strengths of this study are its large sample size and the **extensive** analyses that we have undertaken.
(page 16, lines 237-238)*

11. Fig 4: great figure that summarises a lot of data. It wasn't immediately obvious what the dots represented (I assume all the dots in the orange track are the 955 BW associated CpGs?).

Thank you. The dots represent CpG-specific associations ($-\log_{10}(P)$), each dot represents a CpG. We have clarified this in the legend of figure 4 (page 22):

*Results are presented as CpG-specific associations ($-\log_{10}(P)$, **each dot represents a CpG**), by genomic position, per chromosome.*

Reviewer #3 (Remarks to the Author):

Summary: This is a clear, well written manuscript with extensive analytical depth. The authors evaluate the association between birthweight and DNA methylation in the PACE consortium, including 24 independent cohorts in this meta-analysis. The expertise and experience in this group is well suited to tackle this important research question. They hypothesize that birthweight may impact later life health outcomes through epigenetic mechanisms and compare their results with other findings from PACE, including smoking during pregnancy and pre-pregnancy BMI studies. This paper extends from earlier PACE work by including more cohorts, incorporating mendelian randomization, additional public databases, and more thorough analyses among study participants across life stage. All comments below are minor and for the authors' consideration to potentially improve clarity for the reader.

The statistical analyses are all appropriate, and replicate methods used in previous work from the PACE consortium, while adding depth to functional and causal approaches. This work is of great interest to the scientific communities interested in early life exposures and long-term health implications, as well as the underlying mechanisms explaining these associations.

1. Title: "Meta-analysis of epigenome-wide association studies in neonates reveals widespread differential DNA methylation associated with birthweight." The title does not reflect the longitudinal aspects of this work

which I find to be rather innovative, despite limited conclusive findings. Authors could consider minor modification to the title to reflect the approach.

We thank the reviewer for this suggestion. We feel that the main focus of the manuscript is the cross-sectional analyses in neonates. Importantly, we have relatively little data for the longitudinal analyses which make the results less robust. Although we think there is definite value in the longitudinal aspects of our work, we feel that the main message of the paper is captured in the current title.

2. Causal analyses: The authors note the following, starting on line 355:

“This large-scale meta-analysis shows that birthweight is associated with widespread differences in DNA methylation. We observed enrichment of birthweight-associated CpGs among sites that have previously been linked to smoking during pregnancy²¹ and pre-pregnancy BMI²², consistent with the hypothesis that epigenetic pathways may underlie the observational associations of those prenatal exposures with birthweight^{17,19,45}.”

-Separately, the authors used MR to explore causal association of methylation at identified CpGs and birthweight as well as later-life health outcomes. How was maternal smoking during pregnancy incorporated into these models? Could they address their statement above in a more direct analysis such as MR by evaluating the causal pathway across maternal smoking \diamond CpGs \diamond birth weight? Or maternal smoking + pre-pregnancy BMI \diamond CpG methylation \diamond birth weight?

Following the comments of reviewers 1 and 2 we have now decided not to present the full MR results; please see our detailed response to reviewer 1, point 1. Therefore, we were not able to explore this causal role of smoking and subsequent mediation in this manuscript.

3. Effect measure modification by race/ethnicity: The authors note that the results were consistent when restricting to just European ancestry study participants (Supp Fig 2, blue track in Fig 4). What do they find when restricted to African ancestry study participants only and/or Hispanic ancestry only? Even though the sample size/statistical power is dramatically reduced (just NEST and Healthy Start for AA cohorts), it would be helpful to indicate whether there is consistent direction of effect for top CpGs. This could be a sensitivity analysis in the supplement. Alternatively, they could test for effect measure modification of the association between methylation and birthweight by race/ethnicity for top 1-5 or so CpGs. And given the imbalance in sample size in European vs. non European ancestry, they could consider a random sample of the # of AA subjects among the Europeans so the analyses are not swayed by sample size alone. E.g 1000

Europeans and 1000 AAs in one of the proposed sensitivity analyses. Overall, these suggestions are a broad request for the authors to add some analytical component to further address whether the findings are consistent across race/ethnicity, rather than potentially sweep that aspect under a sample size rug, if you will. This would be beneficial for the research community looking to replicate the presented approaches.

The ethnic-specificity of DNA methylation is potentially interesting, but given the much smaller sample size of the African American and Hispanic samples, we did not primarily run meta-analyses specifically in these ethnic subgroups and do not feel that our data have sufficient power to explore effect modification by ethnicity. In response to this and reviewer 1, we have now explored the correlations between our main analyses results (with all ethnic groups included) and those restricted to Europeans, African Americans and Hispanics (please see response to reviewer 1, point 3 above for full details).

4. Consistency of effects across life stage: Starting on line 296: "In childhood (2-13y; 2,756 children from 10 studies), adolescence (16-18y; 2,906 adolescents from 6 studies) and adulthood (30-45y; 1,616 adults from 3 studies), we observed 91, 51 and 44 of the 955 CpGs, respectively, to be nominally associated with birthweight ($p < 0.05$), with consistent directions of association. Eleven CpGs showed differential methylation across all 4 age periods." -Was the direction of effect for these CpGs consistent?

We have now adjusted the text, emphasising that all directions of association were consistent and adding correlation coefficients to the Results section. (please find below the text in red for the relevant changes, because of the many textual changes we decided to only upload a clean manuscript file without track changes)

In childhood, adolescence and adulthood, we observed 91, 51 and 44 of the 955 CpGs to be nominally associated with birthweight ($p < 0.05$). All these CpGs showed consistent directions of association. (page 11, lines 108-110)

We also adjusted the text in the Discussion regarding this consistency across life stages, please find details in our response to reviewer 1, point 7.

5. MR, starting on line 338: " To explore their causal associations with birthweight and 139 complex later-life outcomes (Supplementary Table 12), we used 135 local methylation quantitative trait loci (cis-mQTLs), genetic variants associated with methylation levels, using a publicly available mQTL database⁴³ as instrumental variables for 127 of the 955 birthweight-associated CpGs in two-sample MR. For the remaining CpGs (i.e. 828 [87%]) no genetic instrumental variables could be identified in the publicly available mQTL database. Hence, we could not conduct MR for those CpGs." --Don't these cohorts also have GWAS data? Why were genetic variants not used in the MR approaches?

Several, though not all, of the cohorts in the analysis have GWAS data available. To identify mQTLs for the birthweight-associated CpGs using the available genetic data, we would have to run a GWAS for each of the 955 CpGs in each PACE study with genetic data and then meta-analyse the results for each CpG. However, as selecting the CpGs to undertake a GWAS within the same data is likely to result in some over-fitting of the data, ideally we would want to run GWAS of EWAS (i.e. of all 450K sites) in each study (and if possible include some additional independent, of PACE, studies). We feel that either is beyond the scope of this current project. As noted previously we have now removed the main MR analyses and results from this paper (see response to reviewer 1, point 1, for full details).

- 6. Confounding by cell type, lines 397-401: The authors addressed the issue of potential confounding by cell type by adjusting for cell type composition in their analyses. They note that they used the adult reference panel which had now been improved by the availability of cord blood reference panels. They conduct a sub analysis using one of the updated cord panels but did not rerun the entire EWAS with this updated reference. This seems appropriate and adequately addresses the issue, given the extensive analysis involved in these types of meta-analyses.**

Relevant excerpt: "DNA methylation varies 397 between leukocyte subtypes⁴⁹ and we used an adult whole blood reference to correct for this in our main analyses^{50,51}, as our study-specific analyses were completed before the widespread availability of specific cord blood reference datasets^{52,53}. However, we observed very similar findings in two studies (Generation R and GECKO) when we compared the results with those using one of the currently available cord blood references⁵²."

We thank the reviewer for this comment and for his/her understanding of the complexity of large-scale consortium meta-analyses.

- 7. Consideration of DAGs: Figure 1 displays hypothetical paths. Given the number of epidemiologists involved in this consortium, modification of the figure to display a proper DAG could be given some consideration, including potential confounders. This could be provided in a supplemental figure rather than replacing Figure 1, which provides the best conceptual overview for the paper. I am not suggesting a modification to their analyses based on what the DAG looks like, but to consider clearer integration of some epidemiologic methods in this type of omics work, either here or in the future.**

Figure 1 shows the conceptual framework for this study. We agree that DAGs can be useful in a lot of situations, but they were not used *a priori* in this project. In response to reviewer 1, point 2 we now discuss residual confounding more clearly across the manuscript. We do not feel that the *post-hoc* addition of a DAG would help to clarify this

paper, though we thank the reviewer for these thoughtful comments and do agree that the understanding and application of epidemiological principles will be an important feature of this type of research in the future.

- 8. Gambia study: The authors acknowledge the study from the Gambia but this is not included as one of the cohorts in the meta-analysis (e.g., not listed in Table 1). It appears only relevant to the MR analyses which appears to be online publicly accessible data. Inclusion of EWAS data from this cohort, if available, would have added some valuable diversity to the study population. In the acknowledgements, it looks like the cohort data was involved.**

The Gambia study does not have DNA methylation data available at birth and therefore was not included in our main analyses. It was used in the analyses of persistence of associations in childhood as well as for the association between DNA methylation and gene expression. In the revised manuscript we have clarified that some studies included in those analyses are not in the main EWAS. We also now acknowledge in the Discussion that it is possible that inclusion of the Gambia (a non-European population) study in the childhood EWAS (but not the main birth EWAS) might impact on those results. However, we note that participants from the Gambia EWAS make up just 7% of the total sample used for the child follow-up.

Methods

*Analyses of the associations with DNA methylation in blood collected in childhood, adolescence and adulthood followed the same covariable adjustment and methods as for the main analyses ($p < 5.2 \times 10^{-5}$ for 955 tests). All participants and studies in these analyses at older ages had not been included in the main meta-analysis in neonatal blood, except for ALSPAC (N=633 in neonatal analyses, N=605 in childhood and N=526 in adolescence), CHAMACOS (N= 283 in neonatal analyses and N=191 in childhood) and Generation R (N=717 in neonatal analyses and N=372 in childhood). Characteristics are shown in study-specific **Supplementary Methods** and **Supplementary Table 1B**. (page 26, lines 420-426)*

Discussion

It is possible that inclusion of the Gambia study in the childhood EWAS (which was the only non-European study in these analyses and was not included in the main meta-analyses with neonatal blood) might have impacted these results, although this study made up just 7% of the total child follow-up sample. (page 15, lines 205-208)

REVIEWERS' COMMENTS:

Reviewer #1 (Remarks to the Author):

Kupers et al. have responded well to my concerns regarding their paper and this is to be commended. However, their clear acknowledgement of all these issues, re causality, mechanism etc, highlighted in the review, does weaken the significance of their findings.

- Too little support to draw firm conclusions on causality/ lack of causality findings via MR
- Significant, as suspected, overlap with a larger set of smoking--related CpGs identified
- Agreement that the biological plausibility of the hypothesis proposed is not directly tested in the study, etc

Furthermore, I am afraid that I disagree with the authors in regard to the inclusion of the 41 CpGs that are clearly documented as cross--reactive/multimapping probes. These should be excluded from any downstream analysis. The suggested Gap Hunting analysis was not to identify these, where there is a obvious methodological reason for probe bias, incorrect methylation scoring, and necessary exclusion – which may be too subtle to show multimodality in all cases. Moreover, there is a clear strong representation of these confounded probes in their results (41/ 955). Gap hunting was to identify any further CpGs that may be flagged beyond these well--known cross--reactive probes – particularly in this multi--population analysis.

Reviewer #2 (Remarks to the Author):

The authors have responded thoughtfully to my comments (and those of the other reviewers). The difficulties in defining causal relationships are now well expressed. Though this has inevitably limited the conviction with which the biological inferences can be communicated, it is also a valuable corrective to see these limitations made explicit in ways that have been all too uncommon in this area.

I did find the "non treatment" of the MR analysis a bit odd. I agree with their decision to remove the MR analysis, and felt this could have been handled by a paragraph in discussion, rather than introducing MR as an aim in the intro, and then using a page of results to say that the analysis promised in the introduction turned out not to be possible.

Reviewer #3 (Remarks to the Author):

The authors have sufficiently addressed all comments provided in the initial review of this manuscript.

Reviewer #1 (Remarks to the Author):

Kupers et al. have responded well to my concerns regarding their paper and this is to be commended. However, their clear acknowledgement of all these issues, re causality, mechanism etc, highlighted in the review, does weaken the significance of their findings.

- Too little support to draw firm conclusions on causality/ lack of causality findings via MR
- Significant, as suspected, overlap with a larger set of smoking--related CpGs identified
- Agreement that the biological plausibility of the hypothesis proposed is not directly tested in the study, etc

Furthermore, I am afraid that I disagree with the authors in regard to the inclusion of the 41 CpGs that are clearly documented as cross--reactive/multimapping probes. These should be excluded from any downstream analysis. The suggested Gap Hunting analysis was not to identify these, where there is a obvious methodological reason for probe bias, incorrect methylation scoring, and necessary exclusion – which may be too subtle to show multimodality in all cases. Moreover, there is a clear strong representation of these confounded probes in their results (41/ 955). Gap hunting was to identify any further CpGs that may be flagged beyond these well--known cross--reactive probes – particularly in this multi--population analysis.

Response: According to the reviewer's input, we have now removed the cross-reactive probes from all results and downstream analyses. Removal of the cross-reactive probes resulted in very minor changes in the (decimals of) results and therefore did not change the conclusions of our manuscript in any way. However, we reran all analyses after removal of the cross-reactive probes, including the Gene Ontology enrichment analyses. For these GO and KEGG analyses we used the most recent underlying reference data (accessed in November 2018), which has substantially changed compared to when we accessed this in 2017. This updated GO analysis resulted in no enriched GO or KEGG pathways and we therefore removed the supplementary table and only mention the methods and results in the text.

Please find below the major textual changes accompanying this removal of cross-reactive probes:

Statistical methods (lines 350-352 in the file with track changes): “*We removed CpGs that co-hybridised to alternate sequences (i.e. cross-reactive sites), because we cannot distinguish whether the differential methylation is at the locus that we have reported or at the one that the probe cross-reacts with. We compared the birthweight-related CpGs to lists of CpGs that ~~co-hybridise to alternate sequences (cross-reactive sites)~~ or that are potentially influenced by a SNP (polymorphic sites)^{26,27}.”*

Results > meta-analysis (lines 104-107 in the file with track changes): “*We identified that 41 of the 955 differentially methylated CpGs ~~co-hybridised to alternate sequences (i.e. cross-reactive sites; Supplementary Table 6).~~ For these we cannot distinguish whether the differential methylation is at the locus that we have reported or one that the probe cross-reacts with.*”

Results > meta-analysis (lines 104-107 in the file with track changes): “*The 955-914 birthweight-associated CpGs showed no functional enrichment of ~~126~~ Gene Ontology (GO) terms or Kyoto Encyclopedia of Genes and Genomes (KEGG) terms ~~involved in a range of pathways (FDR<0.05; Supplementary Table 14).~~*

Discussion (lines 276-278 in the file with track changes): “*We ~~emphasise interpretation with caution for~~ removed the CpGs that were flagged as potentially cross-reactive, as the measured methylation levels may represent methylation at either of the potential loci.*”

Reviewer #2 (Remarks to the Author):

The authors have responded thoughtfully to my comments (and those of the other reviewers). The difficulties in defining causal relationships are now well expressed. Though this has inevitably limited the conviction with which the biological inferences can be communicated, it is also a valuable corrective to see these limitations made explicit in ways that have been all too uncommon in this area.

I did find the "non treatment" of the MR analysis a bit odd. I agree with their decision to remove the MR analysis, and felt this could have been handled by a paragraph in discussion, rather than introducing MR as an aim in the intro, and then using a page of results to say that the analysis promised in the introduction turned out not to be possible.

Response: After lengthy consideration, we opted not to include the MR results in the manuscript as we were only able to identify reliable mQTLs (instrumental variables) for a small minority of the birthweight-associated CpGs. This limited our capacity to undertake comprehensive MR for the full data set and may have led to inferences that were not representative. We did however feel it was important to state our intention to undertake this analysis, as it would be appropriate to do so once additional mQTLs have been mapped across the genome.

Reviewer #3 (Remarks to the Author):

The authors have sufficiently addressed all comments provided in the initial review of this manuscript.

Response: Thank you.